# Tuned polymerization of the transcription factor Yan limits off-DNA sequestration to confer context-specific repression

C Matthew Hope[1], Jemma L Webber[2], Sherzod A Tokamov[3], Ilaria Rebay[2,3]*

[1]Department of Biochemistry and Molecular Biophysics, University of Chicago, Chicago, United States; [2]Ben May Department for Cancer Research, University of Chicago, Chicago, United States; [3]Committee on Development, Regeneration, and Stem Cell Biology, University of Chicago, Chicago, United States

**Abstract** During development, transcriptional complexes at enhancers regulate gene expression in complex spatiotemporal patterns. To achieve robust expression without spurious activation, the affinity and specificity of transcription factor–DNA interactions must be precisely balanced. Protein–protein interactions among transcription factors are also critical, yet how their affinities impact enhancer output is not understood. The *Drosophila* transcription factor Yan provides a well-suited model to address this, as its function depends on the coordinated activities of two independent and essential domains: the DNA-binding ETS domain and the self-associating SAM domain. To explore how protein–protein affinity influences Yan function, we engineered mutants that increase SAM affinity over four orders of magnitude. This produced a dramatic subcellular redistribution of Yan into punctate structures, reduced repressive output and compromised survival. Cell-type specification and genetic interaction defects suggest distinct requirements for polymerization in different regulatory decisions. We conclude that tuned protein–protein interactions enable the dynamic spectrum of complexes that are required for proper regulation.
DOI: https://doi.org/10.7554/eLife.37545.001

***For correspondence:**
irebay@uchicago.edu

**Competing interests:** The authors declare that no competing interests exist.

## Introduction

Progressive cell fate acquisition during development requires extraordinary precision in gene expression. Sequence-specific transcription factors (TFs) direct cell fate decisions by binding *cis*-regulatory elements, known as enhancers, in order to activate or repress gene expression (*Davidson and Levine, 2008*; *Spitz and Furlong, 2012*; *Smith and Shilatifard, 2014*). Two types of interactions recruit and organize transcriptional regulatory complexes: protein–DNA interactions and protein–protein interactions. Protein–DNA interactions allow TFs to decipher the sequence information presented by non-coding DNA — namely the number, order, and affinity of TF binding sites — and build appropriate regulatory complexes at the correct genomic locations. Protein–protein interactions among TFs serve multiple purposes. For example, cooperative or antagonistic interactions can respectively increase or decrease TF occupancy, whereas other layers of interaction can recruit cofactors and general regulators of RNA polymerase (*Funnell and Crossley, 2012*). We currently have a poor understanding of how each TF's protein–DNA and protein–protein interaction affinities are tuned to assemble regulatory complexes that have the composition and dynamics needed to shift gene expression patterns as development proceeds.

As eukaryotic genomes encode a small number of TFs relative to the number of genes to be regulated (*Charoensawan et al., 2010*), TFs must be used reiteratively in distinct contexts throughout development. Furthermore, many TFs are organized into large families that are defined by similarity in DNA-binding preference. This presents a complicated specificity problem, as not only must each

TF regulate a particular gene in some cells but not in others, but TFs from the same superfamily, when coexpressed in a cell, may need to regulate distinct sets of target genes.

Recent manipulations of enhancer sequences across multiple organisms have emphasized the importance of low- and medium-affinity protein–DNA interactions in generating specific developmental expression patterns without ectopic induction (*Lorberbaum et al., 2016*; *Crocker et al., 2015*; *Cary et al., 2017*; *Farley et al., 2015*). In addition to sub-optimized protein–DNA interactions, these studies have also noted that the clustering of multiple low-affinity sites (or multiple enhancers) is capable of restoring robust activation while maintaining specificity. Because protein–DNA affinity is held constant, these results suggest that enhancer activity can be profoundly shifted by favoring or dis-favoring cooperative protein–protein interactions among TFs. In support of this idea, the abolition of TF protein–protein interactions decreases TF occupancy in vitro and in vivo, for a wide variety of TF families (*Johnson et al., 1979*; *Lebrecht et al., 2005*; *Morgunova and Taipale, 2017*). In addition, careful dissection of model enhancers across organisms have shown that changing the spacing of TF binding sites can have strong effects on enhancer output (*Thanos and Maniatis, 1995*; *Yuh and Davidson, 1996*; *Swanson et al., 2010*), and the conservation of binding-site motifs is thought to preserve cooperative TF binding (*Kazemian et al., 2013*).

Although clustered TF binding sites almost certainly affect enhancer output by favoring TF protein–protein interactions, direct manipulation of these interactions is rarely attempted because of the inherently greater complexity of tuning a protein–protein interaction interface relative to manipulating DNA sequence. Thus despite their importance, fundamental questions of how protein–protein interaction affinity impacts regulatory dynamics or target gene specificity in vivo remain unanswered. For example, do higher-affinity interactions stabilize transcriptional complexes to increase activating or repressive output? Or, alternatively, do they compromise regulatory dynamics or target gene specificity and thereby dysregulate enhancer output, analogous to what has been found with protein–DNA interactions?

The *Drosophila* E-Twenty-Six (ETS) family TF Yan, also referred to as Anterior open (Aop), provides a well-characterized system that is ideally suited to exploring these ideas because its protein–DNA and protein–protein interactions can be controlled directly. Yan is a conserved transcriptional repressor that works downstream of receptor tyrosine kinase (RTK) signaling to regulate cell fate decisions during embryonic and retinal development (*Lai and Rubin, 1992*). Upon RTK activation, mitogen-activated protein kinase (MAPK) phosphorylates Yan (*O'Neill et al., 1994*). This dismantles repressive complexes that involve Yan and allows their export from the nucleus and subsequent degradation in the cytoplasm, which in turn enables activating TFs and cofactors to access and turn on previously repressed target genes thereby initiating cell fate transitions. If MAPK-mediated phosphorylation of Yan is blocked, Yan remains nuclearly localized, constitutively represses transcription, and blocks the fate switch (*Rebay and Rubin, 1995*). By contrast, insufficient repressive activity involving Yan permits the inappropriate induction of gene expression programs with ensuing ectopic fate specification (*Lai and Rubin, 1992*; *Rogge et al., 1995*).

Two motifs mediate Yan's protein–DNA and protein–protein interactions to organize its functions and dynamics: the ETS DNA-binding domain and the sterile alpha motif (SAM). The ETS domain binds to DNA motifs with a core GGAA/T sequence and also provides a nuclear localization sequence (*Hollenhorst et al., 2011*). An intriguing feature of the Yan SAM is that the isolated domain forms long helical polymers in vitro (*Qiao et al., 2004*). How SAM–SAM interaction affinity shapes the distribution of polymer length to ensure proper regulation of Yan targets in vivo is not well understood. Mutations that completely block SAM–SAM association and limit Yan to a monomeric state abrogate Yan's repressive ability and show strong loss-of-function phenotypes (*Webber et al., 2013*; *Zhang et al., 2010*; *Qiao et al., 2004*). Mutations that restrict Yan to dimers have close-to-wildtype function, although higher-order Yan complexes exist in cells and can influence repressive output at certain target enhancers (*Zhang et al., 2010*). In addition, a recent modeling effort by our group predicted that longer TF polymers are less able to discriminate specific versus non-specific DNA binding sites (*Hope et al., 2017*), whereas dissection of the regulatory syntax of a classic Yan target enhancer suggests that an intricate balance of site affinity and spacing is necessary for proper function (*Lachance et al., 2017*). Together, these results hint that the strength of Yan self-association must be calibrated to balance DNA occupancy, binding site specificity and repressive output.

In this study, we combined computational and experimental strategies to explore how increasing Yan's protein–protein interaction affinity impacts its DNA occupancy and repressive function. First, mathematical modeling uncovered a regime of affinity in which strong SAM–SAM interactions drive the sequestration of Yan complexes away from DNA. To test this prediction, we exploited structural conservation between Yan and its human ortholog TEL/ETV6 in order to identify four residues whose substitution increased Yan SAM affinity over four orders of magnitude. Increasing SAM affinity impaired Yan's repressive activity and function, and strikingly redistributed Yan into punctate structures in the cytoplasm and nucleus. In addition to these global effects, cellular analysis of high-affinity Yan–SAM mutants in the eye uncovered a novel *yan* loss-of-function phenotype, which we propose reveals different requirements for Yan polymerization at different target genes. Taken together, our results suggest that a spectacular range of affinities lies latent within the SAM, and that these affinities must be tuned to preserve transcriptional repression and biological function during development.

## Results

### Modeling Yan DNA occupancy predicts a requirement for tuned protein–protein interaction affinities to limit off-DNA aggregation

To explore the functional consequences of altering TF protein–protein interactions, we first updated our in silico model of Yan occupancy on DNA at equilibrium (*Hope et al., 2017*) to also consider oligomerization off DNA. Briefly, the published model calculates DNA occupancy using four parameters: the strengths of sequence-specific DNA binding, non-specific DNA binding, and polymerization, as well as the concentration of Yan. The model is then parameterized using measured values of these affinities from Yan and other ETS-family TFs (*Qiao et al., 2004*; *De et al., 2014*). Given that isolated SAM domains are sufficient to undergo polymerization in vitro (*Kim et al., 2001*; *Qiao et al., 2004*), it is likely that SAM-mediated self-association of Yan occurs both on and off DNA, and that competition for Yan molecules between the two pools could be a key feature that shapes Yan DNA occupancy, and ultimately repression.

To incorporate this behavior, we reassessed Yan occupancy using the dual approach of calculating 'on-DNA' microstates as before (*Hope et al., 2017*) while also treating the distribution of off-DNA polymers analytically (see 'Materials and methods'). A key assumption of the approach is that smaller-order Yan species can diffuse effectively in order to bind their targets on DNA, but as Yan oligomers increase in size, the inverse relationship between particle size and diffusion rate, as described by the Stokes–Einstein relation (*Berg and von Hippel, 1985*), will eventually prevent diffusion and DNA binding. Although there have been no direct measurements of Yan diffusion rates on a precise polymer-species basis, differences in bulk diffusion rates have been observed in vivo for Yan wildtype, monomers, and dimers (*Zhang et al., 2010*). Therefore, we assume that at some limiting size of Yan polymer, the diffusion rate will be substantially attenuated.

The results of the calculation predict that Yan fractional occupancy will differ substantially when polymerization on- and off-DNA is considered versus when only on-DNA polymerization is allowed. To show this, we calculated fractional occupancy for a single ETS site in a larger element for three values of Yan SAM affinity (*Figure 1A–C*). In the case of wildtype Yan SAM affinity (*Figure 1A*), fractional occupancy at ETS sites was predicted to be almost the same at lower concentrations of Yan, regardless of off-DNA polymerization. However, at higher concentrations of Yan, the calculation that included polymerization off of DNA predicted a drop in occupancy (*Figure 1A*, black solid curve), whereas that in which only on-DNA polymerization was permitted predicted sustained high occupancy (*Figure 1A*, grey dashed curve). This trend held true when SAM affinity was either decreased (*Figure 1B*) or increased (*Figure 1C*), although the concentration at which the two curves diverged shifted. When SAM affinity was weaker, a greater concentration of Yan was required to achieve substantial fractional occupancy, and vice versa for stronger SAM affinity. Despite these distinctions, all three calculations revealed a regime in which fractional occupancy becomes attenuated by off-DNA polymerization.

To delineate this relationship more fully, we plotted the concentration of Yan necessary to achieve substantial occupancy (50% or greater) at ETS sites across a large spectrum of SAM–SAM affinity from 0 to −14 kcal/mol. The results show that for any given SAM–SAM affinity, there was a

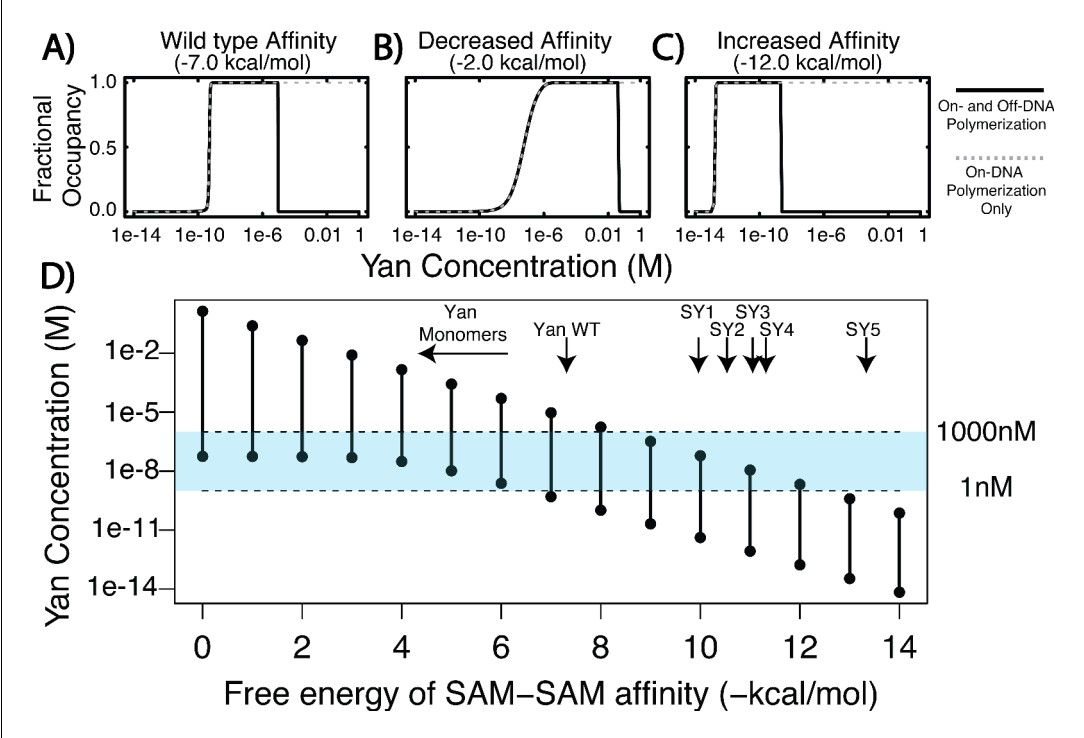

**Figure 1.** Equilibrium modeling shows that increasing polymerization affinity decreases Yan's DNA occupancy. (A–C) Fractional occupancy of Yan for given values of self-association as a function of concentration. Solid lines, polymerization both on and off DNA; dashed lines, on-DNA polymerization only. (D) Concentration ranges of 50% or greater Yan fractional occupancy, as a function of SAM–SAM affinity (black vertical bars). As protein–protein interaction affinity increases, the model predicts that Yan occupancy moves outside the physiologically relevant range (blue shaded area). Arrows point to wildtype affinity (*Qiao et al., 2004*) or to measured SY affinities (this study). The weak affinity of Yan monomers did not permit precise $K_d$ measurement (*Qiao et al., 2004*), but the maximum range of affinity is noted by the beginning of the arrow.

DOI: https://doi.org/10.7554/eLife.37545.002

The following source data is available for figure 1:

**Source data 1.** Increasing the size of species capable of binding DNA does not appreciably alter the output of model calculations.
DOI: https://doi.org/10.7554/eLife.37545.004

defined range of Yan concentration capable of producing substantial occupancy (*Figure 1D*, black vertical bars), and that as SAM–SAM affinity increased, this range narrowed and shifted to a progressively lower Yan concentration. For example, low values of self-association affinity (0 to −5 kcal/mol) produced occupancy only at the high end of the physiological Yan concentration range, whereas increased self-association strength (−10 to −14 kcal/mol) permitted occupancy only at sub-physiological Yan concentrations (*Figure 1D*, blue shading). Thus in contrast to the previous model's prediction that increasing Yan concentration will drive saturated DNA binding across the SAM–SAM affinity range (*Hope et al., 2017*), the incorporation of off-DNA polymerization suggests this will only occur under regimes of moderate self-association where adequate pools of lower-order Yan species remain available to bind DNA.

For simplicity in these calculations, we considered only Yan monomers to be capable of nucleating DNA-bound complexes. To test how this assumption impacted the results, we performed control calculations in which species up to trimers were allowed to search and bind DNA. Similar trends were revealed (*Figure 1—source data 1*). Providing further validation to our assumption, prior work has found that Yan monomers show genome-wide chromatin occupancy patterns that are similar to those of wildtype polymerization-competent Yan (*Webber et al., 2013*), although repressive function is compromised. In addition, in vitro studies have shown that monomerizing mutations in the SAM domain, which is outside of the DNA-binding ETS domain, do not alter Yan's DNA binding ability (*Qiao et al., 2004*). Together, these data suggest that smaller-order Yan species are likely to nucleate DNA-bound complexes in vivo, as we have modeled in silico.

## A screen for Yan mutants that increase SAM-SAM affinity

Our model predicted that a wide range of polymerization affinities can support occupancy at target enhancers, but that extreme values of SAM–SAM affinity are likely to preclude the formation of functional regulatory complexes. Previous experimental work confirms this at the low end of the affinity spectrum, as mutations that prevent the SAM–SAM interaction in Yan produce strong loss-of-function phenotypes owing to impaired regulation of target gene expression (*Webber et al., 2013*). However the second prediction of the tuning hypothesis, that increased SAM–SAM affinity will also be deleterious to Yan function, remains untested.

To engineer Yan mutants with increased affinity, we capitalized on the three orders of magnitude difference that has been measured for the Yan SAM–SAM interaction versus that of its human counterpart TEL (*Kim et al., 2001*; *Qiao et al., 2004*). We reasoned that of the 22 residues at the SAM–SAM interface in the TEL crystal structure, the nine that were divergent between TEL and Yan were probably responsible for this affinity difference (*Figure 2A*). Four of the nine clustered in a small region at the periphery of the SAM–SAM interface, and three of these made salt-bridge contacts that span that interface. Mutating a subset of these residues had previously been shown to decrease TEL's polymerization affinity in vitro and to block the transformation ability of TEL oncogenic fusions in cultured cells (*Cetinbas et al., 2013*). We therefore created all possible combinations of the four mutations Yan R92K, A93E, G96R, and H97Y.

To screen these fifteen Yan-to-TEL (YT) SAM mutants, we adapted a native gel shift assay previously used to compare the polymerization strengths of human SAM domains (*Knight et al., 2011*). Purifying SAMs fused to a super-negative GFP tag (*Lawrence et al., 2007*), which imparts a standard amount of charge to each SAM molecule, allows separation by polymer size and direct visualization on a native PAGE gel (schematized in *Figure 2B*). We included two controls as reference points to mark the lower and upper limits of polymerization: an unequal mixture of two monomeric forms of TEL SAM (TEL A93E and TEL V112E) (*Kim et al., 2001*) to indicate the mobility of monomers and dimers, and the wildtype TEL SAM to show the limited migration of higher-order polymers. The wild type Yan SAM domain ran primarily as monomers and dimers, with some high-order species, consistent with its low µM affinity (*Figure 2C*; *Qiao et al., 2004*).

The mobility shifts observed in the negGFP native gel assay identified Yan SAM variants that matched and exceeded the polymerization strength of TEL SAM (*Figure 2C*). Although two individual mutants, R92K and G96R, modulated mobility on their own (YT7 and YT8, respectively), multiple mutations were required for greater shifts, with simultaneous mutation of all four residues (YT15) surpassing the extent of polymerization by TEL SAM. Three of the mutant SAMs, designated YT1–3, migrated with mobilities that were indistinguishable from that of the wildtype Yan SAM. The next five (YT4–8) shifted mobility toward the dimer form, whereas YT9–11 exhibited a gradation of reduced mobility. Finally, YT12–14 showed mobility indistinguishable from that of TEL SAM, and YT15 migrated even more slowly.

To measure these affinities quantitatively, we turned to pair-wise Surface Plasmon Resonance (SPR) spectroscopy. We selected five mutants whose native gel mobility suggested that they spanned the full affinity range: YT8, YT10, YT12, YT13, and YT15, which we refer to as SuperYan (SY) 1 – 5, respectively. Following the published strategy (*Qiao et al., 2004*), we blocked higher-order polymerization by introducing the monomerizing mutations, A86D and V105R, into each SY construct and then measured the affinity between each SY monomer pair; native gels confirmed that each protein ran as a monomer alone (*Figure 3—figure supplement 1*). The $K_d$s measured by SPR followed the rank order predicted by the native gel shift assay and revealed that four orders of magnitude separate the binding affinities of wildtype Yan and SY5 (*Figure 3A–C* and *Figure 3—figure supplement 2*). Our calculation of the wildtype Yan affinity as 5.6 uM closely matched the previously measured value of 7 uM (*Qiao et al., 2004*), validating the quality of our fusion proteins. In addition, our measurements of 6.7 nM for SY4 and 0.13 nM for SY5 coincide with their migration relative to TEL (2 nM, *Kim et al., 2001*) on the native gels (*Figure 2C*). We also noted non-linearity between mobility on the native gel and affinity, as the rather modest mobility shift of SY1 (YT8) relative to that of the wildtype Yan SAM coincided with a two orders of magnitude decrease in $K_d$. For comparison purposes, the measured values of wild type and SY1–5 in terms of Gibbs free energy are noted on *Figure 1D*.

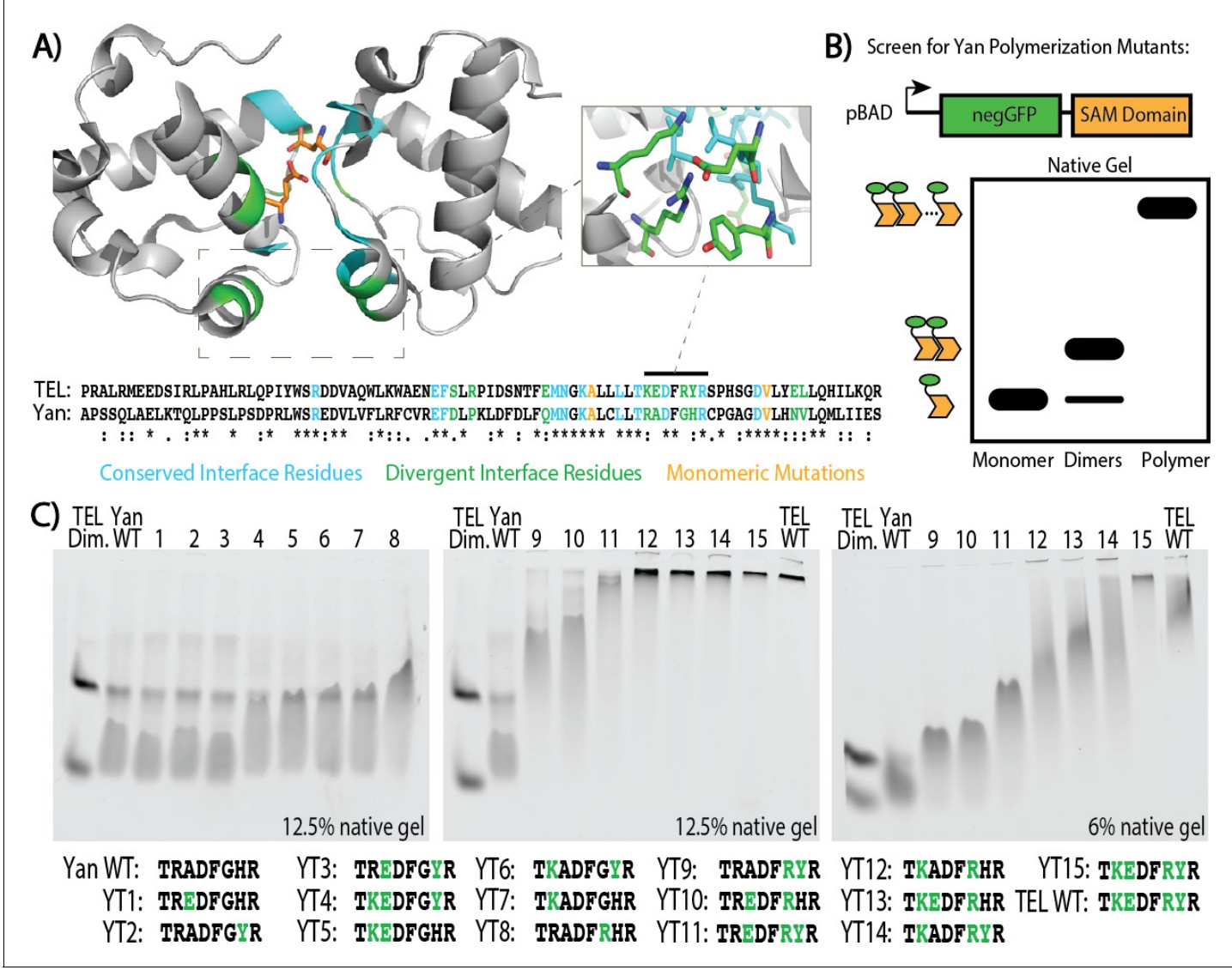

**Figure 2.** Mutagenesis of the SAM–SAM interface increases Yan polymerization affinity. (A) Cartoon representation of the TEL SAM crystal structure (PDB: 1JI7, *Kim et al., 2001*) with the primary sequence alignment of Yan and TEL SAMs below. Residues at the interface are emphasized, with conserved residues in blue, divergent residues in green, and previously published monomerizing mutations in orange (A61D and V80E depicted). Sequence alignment shows conserved residues (*), conservative substitutions (:), and semi-conservative substitutions (.) Inset shows TEL residues corresponding to Yan R92, A93, G96, and H97. (B) Schematic of negGFP Native Gel Assay showing discrimination between monomers, dimers, and higher-order polymers. (C) Native gels showing the mobility of all 15 possible combinations of the four mutations Yan R92K, A93E, G96R, and H97Y relative to wildtype (WT), and arranged in order of increasing polymerization. Sequences are listed below with green font highlighting mutated residues. As reference points, the left-most lane in each gel contains an unequal mixture of TEL monomer species to mark the mobility of monomers and dimers whereas the wildtype TEL SAM shows the mobility of a high-affinity polymer in the right-most lane. YT9-15 were run on both 12.5% and 6% gels to maximize the discrimination of mobility differences.

DOI: https://doi.org/10.7554/eLife.37545.005

To confirm that increased SAM–SAM affinity could promote self-association between full-length Yan proteins, we assessed co-immunoprecipation of FLAG and HA-tagged constructs expressed in *Drosophila* S2 cells. Under stringent conditions in which the wildtype Yan–Yan interaction was not detectable, SY1–5 showed increasingly robust co-immunoprecipation (*Figure 3D*), which was roughly proportional with their measured affinities (*Figure 2C*). To assess whether the SuperYan mutants can still interact with the wildtype Yan SAM interface, we also performed pair-wise co-immunoprecipitations between Yan wildtype and SY5, using the same stringent conditions as

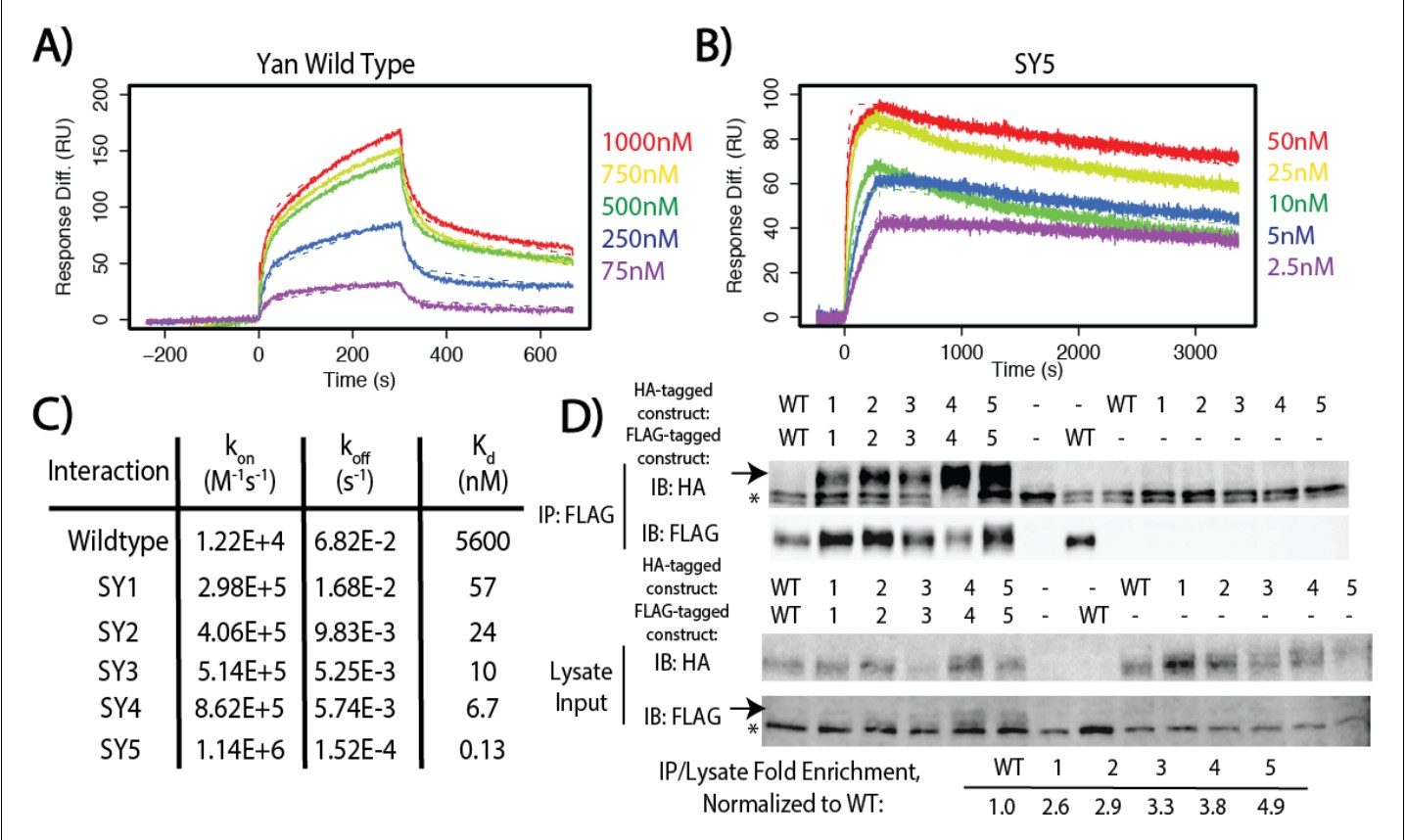

**Figure 3.** SuperYan mutations increase SAM–SAM affinity in vitro and full-length self-association in vivo. (**A**) Representative sensogram of the wildtype Yan SAM interaction, via SPR. Concentrations of analyte are color-coded and listed to the right, and the kinetic model is represented by dashed lines. (**B**) Sensogram of the SY5 interaction, demonstrating stronger binding affinity. (**C**) Table of parameters obtained from kinetic fitting of sensograms for Yan and SY1–5 interactions. Two technical replicates were taken for each interaction to calculate parameters. (**D**) Co-immunoprecipitation of Yan and SY1–5 constructs. IPs are shown in the top panel, whereas lysate inputs are shown in the bottom panel. FLAG-tagged Yan constructs were immunoprecipitated and both HA-tagged and FLAG-tagged proteins were immunoblotted. Arrows mark the mobility of the Yan constructs, whereas asterisks mark unreduced IgG in the IP and a non-specific FLAG-reactive species in the lysate, respectively. Fold enrichment of IP to lysate relative to wildtype is noted.

DOI: https://doi.org/10.7554/eLife.37545.006

The following figure supplements are available for figure 3:

**Figure supplement 1.** Yan A86D and V105R mutations can monomerize SuperYan, thereby allowing measurement of affinity.
DOI: https://doi.org/10.7554/eLife.37545.007

**Figure supplement 2.** SuperYan mutations increase SAM–SAM affinity, as measured by SPR.
DOI: https://doi.org/10.7554/eLife.37545.008

**Figure supplement 3.** SuperYan mutations maintain interaction with wildtype Yan SAM interface.
DOI: https://doi.org/10.7554/eLife.37545.009

those used for *Figure 3D*. We observed robust interactions in both directions (*Figure 3—figure supplement 3*, lanes 6 and 7). We note that co-immunoprecipitation of SY5 by Yan (lane 6) was more robust than the converse (lane 7), presumably because SY5–HA molecules interact homotypically with one another, in addition to interacting heterotypically with Yan-FLAG on beads, resulting in a net increase in the number of SY5–HA molecules that are pulled down. Taken together, our experiments establish SY1–5 as a suite of Yan mutants that span a wide range of strong SAM–SAM affinities, which is ideal for assessing how increased protein–protein interaction affinity impacts TF function.

## Increased SAM–SAM affinity compromises transcriptional repression and reduces Yan function in vivo

To assess the impact of increased SAM–SAM affinity on Yan function, we used CRISPR/Cas9-mediated mutagenesis to generate *yan* alleles carrying the three strongest mutants (SY3–5) and compared their phenotypes relative to wildtype and *yan* null backgrounds; we refer to these three alleles, *yan^SY3^*, *yan^SY4^* and *yan^SY5^*, collectively as the SuperYan (SY) mutants. All three behaved as recessive alleles, as no phenotypes were noted in heterozygotes. As a primary test of genetic fitness, we compared the proportion of eclosed *yan^SY^* homozygotes to their heterozygous siblings. In contrast to *yan* null homozygotes, 100% of which die as embryos (*Webber et al., 2013*), all three *yan^SY^* homozygotes were viable. However significant deviations from Mendelian expectations were observed, with *yan^SY5^* the least fit (*Figure 4A*). The viability of the *yan^SY5^* allele in trans to a *yan* null allele also showed strong lethality compared to wildtype (*Figure 4A*).

To determine when the *yan^SY^* homozygotes were dying, we measured embryonic lethality. Lethality increased from 8% in the wildtype control to approximately 25% in *yan^SY3^*, 30% in *yan^SY4^*, and

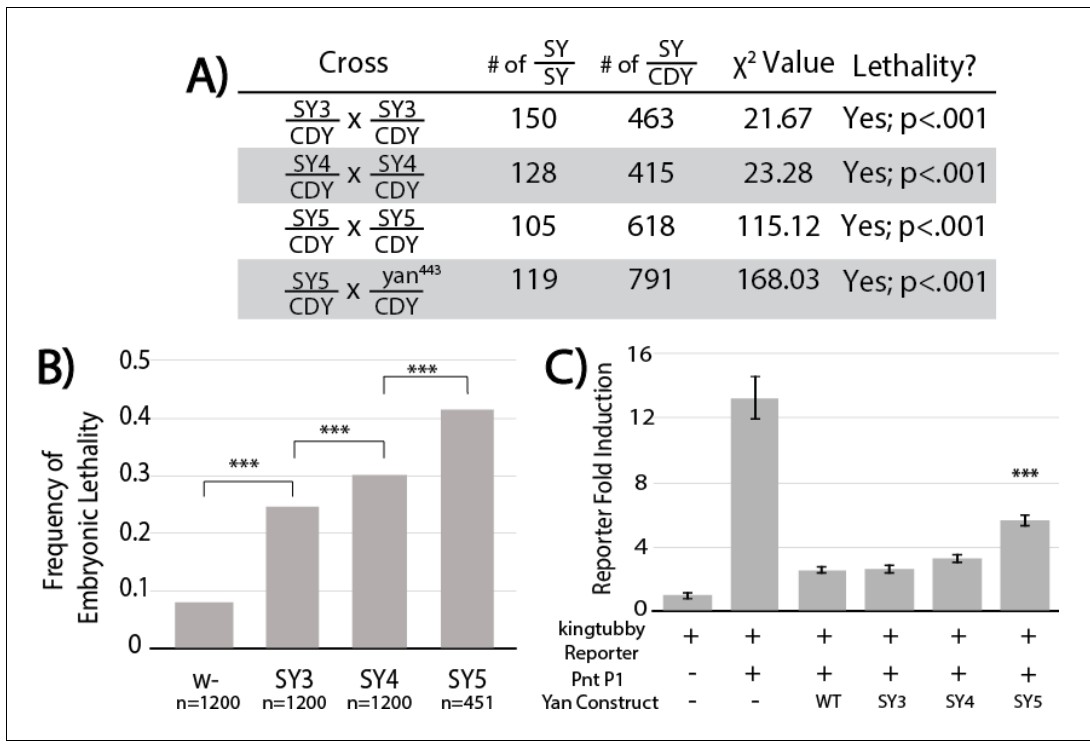

**Figure 4.** SuperYan mutations behave as loss-of-function alleles, and show compromised transcriptional repression. (**A**) The reduced recovery of adults homozygous for *yan^SY3–5^* indicates reduced fitness. Reduced fitness was also observed when *yan^SY5^* was placed in trans to a *yan* null allele, *yan^443^*. CDY, CyO,Dfd > YFP. Significance was calculated via Pearson's chi-square test with the expected Mendelian ratio as the null hypothesis. (**B**) Embryonic lethality of *yan^SY3–5^* homozygotes, as compared to *w*- control, measured by failure to hatch into larvae. Significance was calculated via pair-wise chi-square tests. (**C**) Transcription assays using a Yan/Pnt-responsive transcriptional reporter from *king tubby* show impaired repressive function with SY5. Error bars represent SEM from six biological replicates. Significance was calculated via pair-wise Student T-tests between Yan wildtype and mutants, with Bonferroni correction. ***, p<0.001.
DOI: https://doi.org/10.7554/eLife.37545.010

The following figure supplements are available for figure 4:

**Figure supplement 1.** Overexpressed SuperYan has reduced activity relative to wildtype Yan, but normally localizes to the nucleus.
DOI: https://doi.org/10.7554/eLife.37545.011

**Figure supplement 2.** SuperYan mutants are responsive to RasV12 and properly localized in S2 cells.
DOI: https://doi.org/10.7554/eLife.37545.012

40% in $yan^{SY5}$ (**Figure 4B**), suggesting that the reduced viability (**Figure 4A**) results primarily, but not entirely, from embryonic lethality. Focusing on $yan^{SY4}$ and $yan^{SY5}$, defects in the anterior portion of the head cuticle were noted in the dead embryos, consistent with the classic *yan* 'anterior-open' loss-of-function phenotype (**Nusslein-Volhard et al., 1984**; **Rogge et al., 1995**). Finally, as discussed further below, SuperYan animals that survived to adulthood eclosed with rough eyes, consistent with Yan's known roles in eye development (**Lai and Rubin, 1992**).

Confirming the hypomorphic nature of the $yan^{SY}$ mutants, we showed that their overexpression was less deleterious than the overexpression of wildtype Yan. Thus, GMR–Gal4-driven eye-specific expression of UAS–Yan and UAS–SY1–5 transgenes revealed an inverse correlation between SAM affinity and the severity of the disruption in external eye morphology (**Figure 4—figure supplement 1A–G**). The trend was most striking when driving two copies of the transgenes: whereas eye pigmentation and morphology were virtually obliterated by wildtype Yan overexpression, quasi-wildtype pigmentation and morphology were maintained in animals overexpressing SY5 (**Figure 4— figure supplement 1H–M**). Comparable levels and nuclear localization of the different Yan proteins (**Figure 4—figure supplement 1P**-AB) suggested that impaired activity was responsible for the phenotypic differences.

To test the function of the SY with respect to transcription, we compared repressive activity in transfected cultured cells using reporters generated from regions bound by Yan in vivo (**Webber et al., 2013**). Although the repressive outputs of SY3 and SY4 were statistically indistinguishable from that of wildtype Yan, SY5 showed a reduced ability to repress (**Figure 4C**). Subcellular distribution and signaling responsiveness were indistinguishable from those of the wildtype (**Figure 4—figure supplement 2A–B**). We conclude that impaired repressive ability underlies the milder effects associated with SY overexpression, and by extension, also the loss-of-function phenotypes of $yan^{SY}$ mutants.

## High-affinity SAM–SAM interactions promote nuclear and cytoplasmic Yan aggregation

Motivated by our model's prediction that high SAM–SAM affinity will drive off-DNA polymerization, we compared the sub-cellular localization of SuperYan to that of wildtype Yan in the third instar eye imaginal disc. Striking differences were noted. First, overall Yan levels appeared increased in the SuperYan mutants (**Figure 5A–D**); using SY4 as the example, a side-by-side quantification in mitotic clones confirmed ~50% increases in both total and nuclear levels (**Figure 5F,G**). Second and more striking, the density of punctal structures increased 10 – 20 fold for SuperYan mutants when compared to wildtype Yan, with the phenotype most pronounced in SY5, the highest-affinity SAM mutant (**Figure 5A–E**). Thus in contrast to wildtype control discs in which Yan localized diffusely throughout the nucleus, with only a few puncta detected in nuclei closest to the morphogenetic furrow where Yan levels are highest (**Figure 5A**), in the $yan^{SY}$ mutants, punctal structures were detected throughout the tissue (**Figure 5B–E**). This cellular phenotype matches the modeling predictions of a loss of function resulting from a sequestration mechanism in which the excessive off-DNA aggregation of SuperYan reduces productive binding to target enhancers. Given that the ratio of total to nuclear Yan was comparable in wildtype and $yan^{SY}$ discs, cytoplasmic self-association may primarily slow SY degradation rather than impacting nuclear import kinetics.

To bolster the conclusion that Yan puncta reflect polymerization-driven structures that, in excess, can sequester the transcription factor from its normal functions, we assessed the effect of two different genetic manipulations that are predicted to reduce polymerization. First, because $yan^{SY}$ alleles are fully recessive, we predicted wildtype subcellular localization in the heterozygote. Consistent with expectations, the extent of punctal formation in discs from heterozygous animals carrying one copy of $yan^{SY5}$ and one copy of wildtype *yan* was dramatically lower than that in discs from $yan^{SY5}$ homozygotes (**Figure 5—figure supplement 1B,D**, compare to **Figure 5E**), and showed a modest increase when compared to the frequency of puncta in disks from wildtype homozygotes (**Figure 5—figure supplement 1D**). In light of our detection of Yan–SY co-complexes in cultured cells (**Figure 3—figure supplement 3**), we interpret this result to mean that endogenous Yan can effectively intercalate into SY polymers, and that this capping effect limits polymerization-driven puncta formation. This then restores an adequate pool of Yan for proper transcriptional regulation, resulting in normal development. Second, given that MAPK directly phosphorylates Yan in close proximity to the SAM domain to attenuate repressive function (**Rebay and Rubin, 1995**;

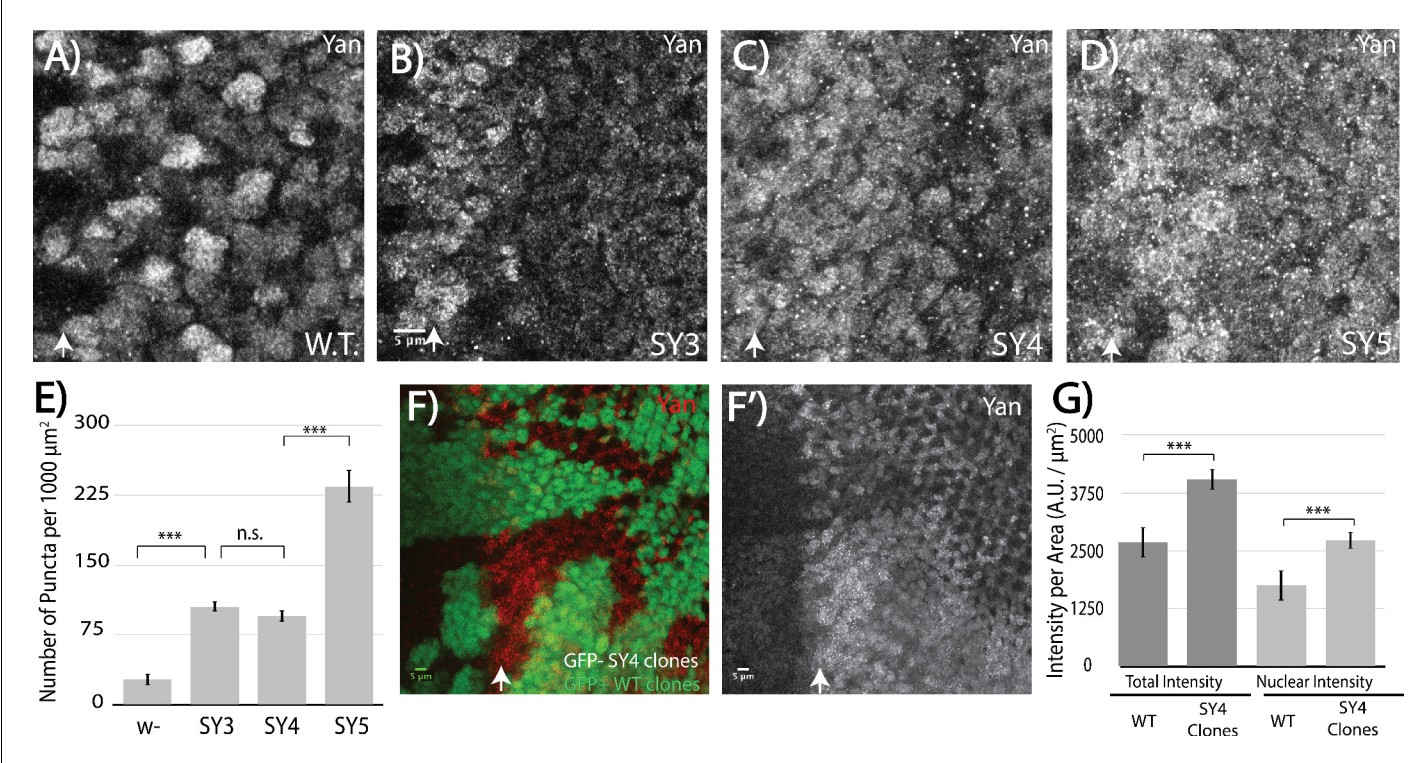

**Figure 5.** High-affinity SAM–SAM interactions increase Yan levels and promote punctual aggregates in both the cytoplasm and the nucleus. (A–D) Immunofluoresence of Yan in wild-type and yan$^{SY3-5}$ third instar eye discs, oriented anterior left. (E) Quantification shows an increased number of puncta per area in SuperYan discs. Error bars represent SEM from n = 6 measurements. Significance was calculated by pair-wise Student T-tests with Bonferroni correction. (F–F') Mitotic clones of yan$^{SY4}$, marked by the absence of GFP, show increased Yan puncta and intensity. (G) Quantification of Yan levels in yan$^{SY4}$ clones relative to wild type. Measurements represent n = 4 pairs of clones, normalized by area, for both total intensity and nuclear intensity. Significance was calculated via pair-wise Student T-tests with Bonferroni correction. Scale bars represent 5 um; white arrows mark the position of the morphogenetic furrow in (A–D) and (F–F'). ***p<0.001; n.s., not significant.

DOI: https://doi.org/10.7554/eLife.37545.013

The following figure supplement is available for figure 5:

**Figure supplement 1.** Levels of RTK signaling influence Yan puncta.

DOI: https://doi.org/10.7554/eLife.37545.014

Tootle et al., 2003), we asked whether overexpression of *rolled* (*rl*) *sevenmaker* (SEM), a hyperactive form of MAPK (*Brunner et al., 1994*), could reduce puncta formation. Quantification showed that SY5 puncta were reduced to levels just above wildtype levels, a dramatic reduction from the yan$^{SY5}$ background without *rl* overexpression (*Figure 5—figure supplement 1C,D*, compare to *Figure 5E*). Taken together, these two results support a model in which the increased polymerization capability of SY drives excessive aggregation into punctal structures, thereby sequestering the repressor from its normal roles.

## Photoreceptor cell-fate specification defects suggest different requirements for Yan polymerization in R7 versus R3/R4

Our results with the SuperYan alleles suggest that the distribution of Yan polymer species is tuned by SAM–SAM affinity and signaling events to allow proper gene regulation. Previous work showing target gene-specific differences in the extent of de-repression when Yan is restricted to monomers (*Zhang et al., 2010*) raises the possibility of context-specific developmental requirements for different distributions of Yan polymer species. Therefore, to assess the impact of increased Yan polymerization on different cell-fate decisions, we turned to photoreceptor specification in the *Drosophila* eye.

As mentioned above, the external eye morphology was visibly disrupted in homozygous adults of all three SuperYan alleles, with the degree of 'roughness' most severe in $yan^{SY5}$ flies (*Figure 6A–D*). Histological sections revealed over-recruitment of photoreceptors, with extra outer photoreceptors more frequent than extra R7's (*Figure 6G–I*, quantification in *Figure 6F*). Thus, ~30% of $yan^{SY4}$ or $yan^{SY5}$ ommatidia recruited at least one extra outer rhabdomere, whereas extra R7-like rhabdomeres were found in less than 10% of ommatidia (*Figure 6F*). When examining position, the extra outer rhabdomeres often appeared near the R3/R4 position. Examination of the expression of Spalt major (Salm), a marker of R3/R4 fate, revealed extra Salm-positive cells in the third instar $yan^{SY5}$ eye disc (*Figure 6L*), supporting the conclusion that many of the ectopic outer rhabdomeres seen in the adult sections belong to R3/4 photoreceptors. Extra Prospero (Pros)-positive nuclei were rare, consistent with the low frequency of extra R7 rhabdomeres in the adult sections (*Figure 6O*).

Although extra photoreceptor recruitment is the predicted phenotype for a *yan* hypomorph, the sensitivity of outer versus inner photoreceptors was strikingly different from that of other characterized hypomorphs where extra R7 rhabdomeres are most common. For example, in $yan^1$ adults, 88% of ommatidia have extra R7s and 27% have extra outer photoreceptors, whereas in a weaker hypomorph, $yan^P$, about 20% of ommatidia have extra R7s but extra outers are rarely detected (*Lai and Rubin, 1992*). Thus in the strongest SY mutant, $yan^{SY5}$, the effect on outer photoreceptors matched that of $yan^1$, but the R7 phenotype was dramatically weaker. These differences suggest that SY can assemble transcriptional complexes that retain significant function with respect to preventing extra R7 fates, but are less able to assemble the regulatory complexes that normally limit R3/R4 fates.

To gain further insight into these differential sensitivities, we sectioned the eyes of $yan^{V105R}$ animals. This allele carries a missense mutation in the SAM that limits Yan to a monomeric state (*Zhang et al., 2010*) and thus represents the other extreme on the SAM affinity scale. Genetically, $yan^{V105R}$ behaves as a strong hypomorph, with occasional adult escapers that have a rough eye phenotype (*Webber et al., 2013*; *Figure 6E*). When sectioned, the pattern of ectopic photoreceptors matched that of $yan^1$, with extra R7-like rhabdomeres in almost 70% of ommatidia and extra outer rhabdomeres in 28% (*Figure 6F,J*). Consistent with the adult eye histological sections, third instar larval eye discs showed an obvious expansion of R7 fate (*Figure 6P*) and a more moderate expansion of R3/4 fate (*Figure 6M*).

## Spalt major is a direct target of Yan, and is preferentially sensitive to increased Yan polymerization

The distinct consequences on R7 versus R3/4 photoreceptor recruitment under regimes of absent or very strong SAM-mediated self-association raised the possibility of a requirement for differently sized Yan-repressive complexes at different target enhancers. Therefore, we sought to connect the observed changes in R7 and R3/4 photoreceptor specification to specific targets of Yan regulation. While *prospero* (*pros*) provided a well-defined R7-relevant target (*Xu et al., 2000*; *Hayashi et al., 2008*), we needed to identify an R3/R4-specific target in order to test this idea. *Spalt major* (*salm*) and *spalt-related* (*salr*) offered logical candidates, given their known role in R3/R4 specification (*Domingos et al., 2004*).

We therefore examined Yan chromatin occupancy patterns at *salm* and *salr* in third instar eye discs, using an unpublished chromatin immunoprecipitation with deep sequencing (ChIP-seq) dataset (Webber, JL and Rebay, I, unpublished). No significant Yan occupancy was detected at *salr* (*Figure 7—figure supplement 1B*), but the Model-based Analysis of Chip-Seq (MACS) peak-calling tool (*Zhang et al., 2008*) identified a high-confidence Yan-bound region just upstream of *salm* (*Figure 7A*; *Figure 7—source data 1*). Examination of Yan ChIP-seq data from stage 11 embryos (*Webber et al., 2013*) did not reveal significant occupancy at either *salm* or *salr*, suggesting that the peak detected at *salm* in the third instar dataset could reflect context-specific regulation relevant to R3/R4 specification (*Figure 7—figure supplement 1A,B*). Curiously, no occupancy was observed at the known *pros* enhancer and MACS did not identify any peaks at the *pros* locus as significant (*Figure 7B* and *Figure 7—figure supplements 1C* and *Figure 7—source data 1*). In contrast to *salm* where we detected context-specific Yan occupancy, more broadly used target genes such as *argos* (*aos*) and *anterior open* (*aop/yan*) showed very similar occupancy profiles in embryos and eye discs (*Figure 7C,D* and *Figure 7—figure supplement 1D,E*).

To determine whether reduced binding of the *salm* region by SY polymers might underlie the ectopic R3/R4 fate induction, we performed ChIP-qPCR (ChIP with quantitative PCR) in both

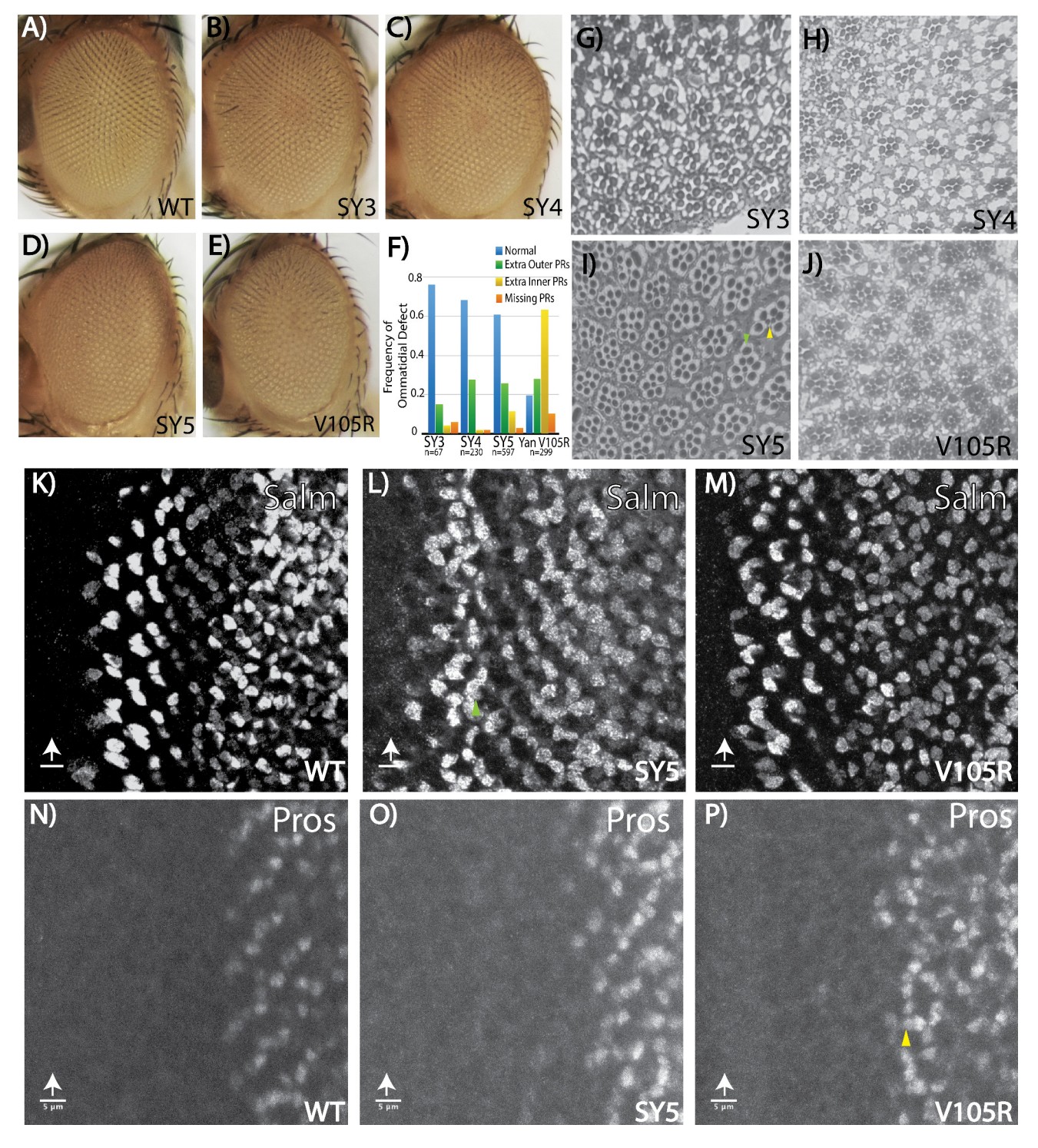

**Figure 6.** Fate-specification defects suggest different requirements for Yan polymerization and repression in R7 versus R3/R4 photoreceptors. (A–E) Adult eye images of indicated genotypes, showing progressive roughening of the eye field for the SuperYan alleles and monomeric Yan V105R. (F) Quantification of sectioning. All genotypes were significantly different from one another, as confirmed by paired Chi-square tests, p<0.001. (G–J) Histological sections of $yan^{SY3-5}$ and $yan^{V105R}$. Green arrows indicate extra outer photoreceptors, yellow arrows indicate extra inner photoreceptors; examples shown in (I). (K–M) The expression of Spalt-major (Salm), a marker of R3/4 cell fate, shows increased numbers of R3/4 cells. (N–P) Expressions of Prospero (Pros), a marker of R7 cell fate, shows no expansion of the R7 lineage in $yan^{SY5}$ discs. White arrows mark the position of the morphogenetic furrow; green and yellow arrows mark examples of extra Salm-positive and Pros-positive cells, respectively. Scale bars represent five microns.
DOI: https://doi.org/10.7554/eLife.37545.015

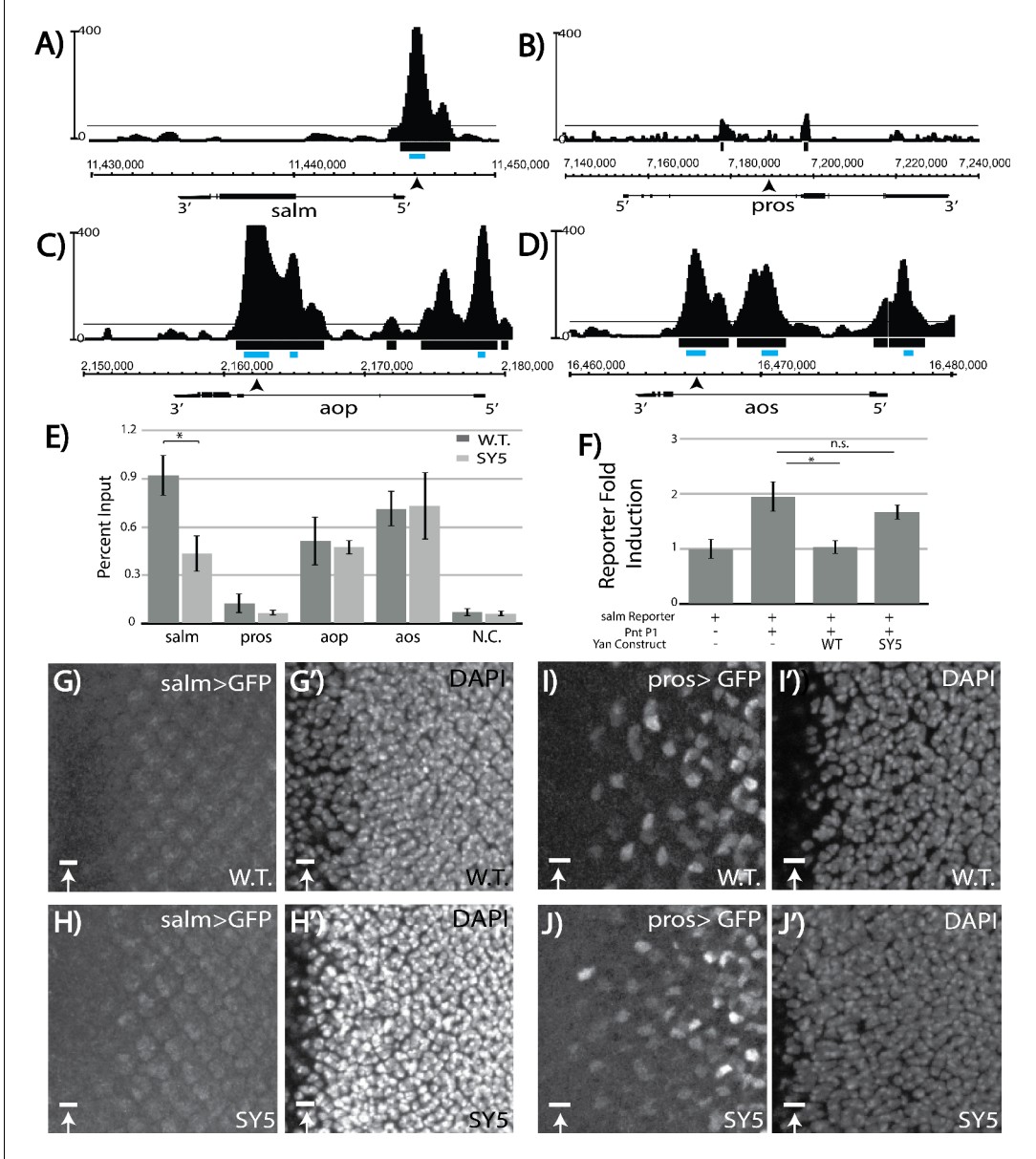

**Figure 7.** Salm is a direct Yan target and is specifically derepressed in *yan*[SY5] mutants. (**A–D**) ChIP-seq tracks of Yan binding in third instar larval eye discs, represented as smoothed tag density with a scale of number of reads per million; alignment was performed to *Drosophila* genome assembly Dm3. For clarity, the longest isoform of each gene is depicted, with 5' and 3' ends noted. Arrows mark the location of the primers used for ChIP-qPCR; gray boxes mark the location of peaks called by MACS (see ***Figure 7—source data 1***). (**A**) *spalt major* locus (*salm*). (**B**) *prospero* locus (*pros*). (**C**) *yan* locus (*aop*). (**D**) *argos* locus (*aos*). (**E**) ChIP-qPCR of wildtype and *yan*[SY5] eye discs. Error bars represent SEM of n = 3 biological replicates. Significance was calculated by Student's T-test *, p<0.05. (**F**) Transcription assays with the *spalt* reporter. Error bars represent SEM of n = 4 biological replicates. Significance was calculated by Student's T-test *, p<0.05; n.s., not significant. (**G–H**) *spalt* reporter-driven GFP expression, in a wildtype (**G**) and SY5 background (**H**). (**I–J**) *pros* reporter-driven GFP expression, in a wildtype (**I**) and SY5 background (**J**). Scale bars represent five microns.
DOI: https://doi.org/10.7554/eLife.37545.016

The following source data and figure supplement are available for figure 7:

**Source data 1.** Genomic coordinates of MACS-called Yan ChIP peaks.
DOI: https://doi.org/10.7554/eLife.37545.018

**Figure supplement 1.** Comparison of Yan ChIP patterns in the eye disc versus embryo reveals eye-specific occupancy at *salm*.
DOI: https://doi.org/10.7554/eLife.37545.017

wildtype and yan^SY5 third instar eye discs. Robust enrichment was seen in the wildtype, validating *salm* as a bone fide Yan-bound gene (*Figure 7E*). Strikingly, SY5 occupancy at *salm* was less than half that measured for wild type Yan (*Figure 7E*), suggesting that Yan binding at regions associated with *salm* is sensitive to the polymerization state of Yan. No significant enrichment was found at the defined *pros* enhancer, consistent with our eye disc ChIP-seq analysis (*Figure 7E*). Failure to detect Yan occupancy at *pros* at the validated enhancer suggests that Yan binding may be either transient and/or occur in only a small number of cells, making the signal too weak to distinguish in a disc-wide analysis.

Because the majority of yan^SY3–5 animals survive to adulthood, most Yan targets must be adequately regulated, and by extension normally bound, by SY. To test this, we extended the comparative ChIP-qPCR analysis to defined regulatory elements in *argos* (*aos*) and *aop* (*yan*) (*Webber et al., 2013*). *aos* is a well-studied direct Yan target and encodes a broadly used negative feedback regulator of EGFR signaling (*Golembo et al., 1996*). Feedback regulation at *aop* is also presumably used throughout development, and MACS-called peaks at *aop* are some of the most enriched genome wide (*Figure 7—source data 1*; *Webber et al., 2013*). Consistent with expectations, equivalent occupancy at both elements was measured in both wildtype and yan^SY5 eye discs. Together, these data suggest that regulatory complexes that are needed for proper occupancy and regulation of *salm* are less likely to form when SAM affinity is strong, whereas adequate regulatory complexes can still assemble at *aos* and *aop*, and perhaps at most target genes.

To confirm the reduced ability of SY5 to bind and repress the *salm* element, we placed it upstream of a luciferase cassette and performed transcription assays in cultured cells. Although the induction of transcription was weaker than the induction observed for the *king tubby* reporter, transcription of the *salm* reporter was induced by Pnt-P1 and repressed by Yan (*Figure 7F*). SY5 showed a significantly reduced ability to repress the *salm* reporter, consistent with its reduced binding to the *salm* promoter region (*Figure 7F*).

We also examined *salm* reporter gene expression and SY5 sensitivity in vivo. In wild type eye discs, the *salm* reporter induced very weak GFP expression with no obvious cell-type specific pattern (*Figure 7G*). However, in the yan^SY5 background, expression in a R3/4-like pattern appeared, suggesting cell-type specific de-repression of the *salm* reporter (*Figure 7H*). As a control, we also examined reporter expression driven by the classic *pros* enhancer and found very similar patterns in wildtype and yan^SY5 discs (*Figure 7I–J*). We conclude that *salm* is a novel direct R3/R4-specific target of Yan in the developing eye disc, and that its regulation is uniquely sensitive to the increased polymerization of Yan.

## Discussion

In this study, we explored how protein–protein interaction affinity shapes the transcriptional behavior and function of the ETS family repressor, Yan. Our results support a loss-of-function by sequestration model in which increased SAM–SAM affinity drives excessive off-DNA self-association, thereby limiting Yan's ability to form effective regulatory complexes at its target enhancers. Focusing on the developing eye, although both very low- and very high-affinity SAM mutants produce *yan* loss-of-function phenotypes, the patterns of the photoreceptor specification defects are distinct. We propose that different distributions of Yan polymeric species regulate different target genes and hence different cell fates, and that tuning SAM–SAM affinity to a middle regime optimizes these requirements.

Our results highlight a fundamental principle for the operation of TFs during development, which is that protein–protein interaction affinity must be tuned through evolution to balance robust occupancy of transcriptional complexes with the ability to assemble those complexes in the first place. A priori it was possible to hypothesize that TF protein–DNA and protein–protein interactions operate independently, with the former required for finding the correct genomic binding sites and the latter stabilizing DNA-bound complexes. This view predicts gain-of-function behavior from mutations that increase protein–protein interaction affinity, with more stable transcriptional complexes exerting stronger effects on RNA polymerase. Instead, we find that the SuperYan (yan^SY) alleles exhibit loss-of-function behaviors that we propose are the result of impaired ability to form functional transcriptional complexes at appropriate DNA sites. Therefore, we hypothesize that the increase in self-association affinity shifts the distribution of Yan polymer species in vivo, thus

depleting (but not abolishing) Yan complexes of the correct size for target gene repression and resulting in the observed hypomorph phenotypes.

Although our experiments focused on a polymerizing TF, we expect that the requirement for tuned protein–protein interaction affinity will extend to many TFs, including those that do not polymerize. Our reasoning relies on three observations. First, although TF polymerization is a relatively uncommon behavior, many TFs operate as obligate dimers (*Funnell and Crossley, 2012*). For these, protein interaction affinity will probably impact specificity and output, just as it does with Yan. Second, even for monomeric TFs, because of the fact that transcriptional regulatory complexes are dominated by combinatorial TF binding during eukaryotic development, heterotypic interactions between different TFs are common and critical to regulatory output. Third, TFs are used iteratively to regulate many different genes throughout development, with each TF participating in different complexes at different target enhancers. In all of these scenarios, we predict that homo- and heterotypic interaction affinities must be tuned to optimize transcriptional regulatory fidelity across enhancers. Mutations that shift this balance thus have the potential to change dramatically the landscape of competition and collaboration between TFs at different enhancers. In extreme cases, excessive affinity will increase off-DNA TF interactions, thus slowing diffusion and hampering the search for target sites on DNA. Thus, although the polymerization of TFs such as Yan may have a dramatic effect on their tuned protein–protein interaction affinity, we expect that many other TFs navigate similar tradeoffs during development.

Recent advances in imaging techniques have highlighted the dynamic nature of transcriptional complex assembly in living cells (*Chen et al., 2014*; *Morisaki et al., 2014*; *Loffreda et al., 2017*; *Paakinaho et al., 2017*). In particular, eukaryotic TFs rely heavily on 3D diffusion to assemble regulatory complexes in the crowded environment of the nucleus, with transient chromatin associations predominating over stable interactions with specific DNA binding sites. Interpreting our results in this context, SAM–SAM interaction affinity must be tuned to maximize success in both the stochastic search for target enhancers and the assembly of appropriately labile yet functional repressive complexes at those elements. Specifically, strong SAM–SAM affinity should increase the proportion of time that a given SuperYan molecule spends in higher-order complexes, thereby limiting the effectiveness of a diffusion-based search. When Yan is limited to a monomeric state, genome-wide chromatin occupancy patterns appear largely unperturbed, at least as visualized in bulk assays from many cells (*Webber et al., 2013*), yet repressive output is compromised (*Zhang et al., 2010*). It is possible that in the absence of SAM-mediated self-association, despite effective diffusion and search behavior, the residence time of DNA-bound Yan molecules is insufficient to form stable transcriptional complexes. Single-cell- and single-molecule-based comparisons of the diffusion, DNA binding, and repressive activity of Yan across the SAM-affinity spectrum will be needed to explore these ideas further.

Despite the difficulties of directly observing polymerized Yan complexes in vivo, the distinct patterns of photoreceptor-specification defects in low- versus high-affinity *yan* mutants provide important hints as to how Yan polymerization may be used to regulate distinct target genes. Focusing on the regulation of target genes relevant to R7 fate specification, and regulation of the R3/R4 determinant *spalt major* (*salm*) (a novel direct Yan target identified in this study), we speculate that there may be context-specific requirements for polymerization. We propose that target genes whose regulation requires lower-order Yan species will be compromised most by increasing SAM–SAM affinity, whereas those that require higher-order species will be most sensitive to reductions in affinity. Our data suggest that genes that are involved in R3/R4 specification, such as *salm*, belong to the first category, whereas those that are important for R7 fate fall into the second. In other words, the shift in mean size distribution toward larger species that are associated with the $yan^{SY}$ alleles results in insufficient smaller-order species to repress R3/4 elements, but sufficient higher-order polymers to regulate R7 elements. The reduced SY occupancy detected at *salm* supports the first assumption, but unfortunately the lack of ChIP sensitivity at *pros* precluded us from testing the second. At the other end of the affinity spectrum, we suggest that the complete loss of SAM–SAM self-association in Yan monomers prevents the formation of the higher-order complexes that are required to repress R7 regulatory elements. For R3/R4 regulation, SAM-independent associations between Yan monomers, perhaps mediated by heterologous interactions with other TFs, must be largely adequate for repression. This model also explains the strong derepression of R7 fate noted in *yan* hypomorphs (*Lai and Rubin, 1992*), as global Yan depletion should preferentially impact regulation at elements

that require higher-order species, because by definition, higher Yan levels are needed to form such complexes.

Below, we discuss two mutually non-exclusive mechanisms that could produce these distinct modes of Yan-mediated regulation: regulatory grammar and signaling environment. In the first case, we envision that different regulatory elements will require Yan oligomers of different lengths to repress transcription, with the specific preferences encoded by the arrangement of TF binding sites. In the second case, we speculate that cell-to-cell signaling could modulate the polymerization of Yan to bias the distribution toward longer or shorter oligomers, as needed in the particular context.

In the case of differential *cis*-regulatory DNA elements, we speculate that the regulatory grammar that produces target-gene-specific differences in Yan occupancy could be as simple as using heterotypic versus homotypic binding sites, which would facilitate smaller and larger Yan complexes, respectively. Previous studies have noted the importance of tandem or multiple ETS sites in Yan regulation of classic target genes such as *even-skipped* (*Webber et al., 2013*; *Lachance et al., 2017*), as well as enrichment for binding motifs of other TFs such as Mothers against Dpp (Mad) (*Webber et al., 2013*), hinting that Yan may participate in both hetero- and homotypic complexes depending on the format of the element in question. In addition to Mad, we speculate that the *Drosophila* AP-1 TFs *kayak* and *jun-related antigen* might also play key roles in setting up heterotypic Yan complexes, as *kayak* has a critical role in establishing R3/4 fate and these factors genetically interact with *yan* and *pnt* in R3/4-dependent planar cell polarity establishment (*Weber et al., 2008*). Given the extensive evidence of physical associations between AP-1 TFs and ETS family factors in mammals (*Li et al., 2000*), these are prime candidates for exploration in the context of the *yan*^SY mutants.

The second mechanism that may enforce the differential regulation of target genes could stem from the different signaling levels across cell types. For example, in R7 precursors, which experience higher levels of RTK signaling than other photoreceptors (*Freeman, 1996*), activated MAPK may more effectively phosphorylate and disrupt SY polymers, thus shifting Yan towards a more wildtype distribution, whereas the lower RTK signaling associated with R3/4 specification would be unable to do so. Consistent with this, overexpression of *rl*^SEM reduced SY puncta, suggesting that RTK signaling can directly modulate polymerization, with the assumption that the prevalence of puncta correlates positively with Yan polymerization. Given that one of the most important MAPK phosphorylation sites in Yan is immediately adjacent to the SAM domain (*Rebay and Rubin, 1995*), we speculate that this site might be properly positioned to either destabilize Yan polymerization or prevent additional polymerization once phosphorylated.

Further experiments will be required to determine the extent to which these two mechanisms contribute to Yan's context-specific regulation of gene expression, but our mathematical models offer some insight into the role of DNA target recognition. Our previous work suggested that the extent of on-DNA polymerization is a trade-off between occupancy and specificity, with dimers best distinguishing specific from non-specific DNA binding sites (*Hope et al., 2017*). As SAM affinity strengthens, the model predicts that higher-order Yan polymers will spread repression into non-specific DNA sites, producing gain-of-function effects. By contrast, the model presented in this study suggests that off-DNA polymerization can provide an effective buffering mechanism to prevent such gain-of-function effects and that in the extreme case of very strong SAM affinity, as exemplified by the SY alleles, it can actually reduce DNA occupancy to drive loss-of-function effects.

Even though the original model that considers only on-DNA polymerization cannot explain the loss-of-function phenotypes of SY mutants, it still provides useful insight into the different repressive complexes that may underlie the R7 versus R3/R4 cell-type specific sensitivity to SY. Specifically, we speculate that Yan may utilize both smaller and larger species to accomplish different goals with respect to repression. Specifically, the accurate recognition and repression of a single enhancer site, such as the one identified at *salm*, may rely on monomers or lower-order oligomers, whereas coordinated repression across a locus with broader patterns of occupancy, such as *aos* or *aop*, may require longer polymers. This idea is consistent with our prior work that demonstrated that although Yan monomers have an overall 'wildtype' DNA occupancy pattern, the density with which peaks clustered across extensively occupied loci such as *aos* and *aop* was reduced. Deeper mechanistic understanding of how polymerization impacts the context-specificity of Yan DNA binding and repression will require both further mathematical modeling to incorporate heterotypic interactions with other TFs and high-resolution genome-wide occupancy studies.

If Yan requires micromolar SAM–SAM affinity for normal function and regulation across a range of signaling environments, then how does its human counterpart TEL operate with nanomolar affinity? One possibility is that TEL evolved a dramatically stronger protein–protein affinity because of the need to form stronger regulatory complexes, either because of the larger size of the mammalian genome or the greater number of co-expressed, competitive ETS factors therein (*Hollenhorst et al., 2011*). Alternatively, TEL could be expressed at much lower levels than Yan, thereby limiting off-DNA aggregation while requiring stronger affinity for effective repression. Finally, heterotypic inter-actions, for example association with the ETS factor FLI1 (*Kwiatkowski et al., 1998*) or post-translational modification, such as phosphorylation by MAPK or sumoylation (*Maki et al., 2004*; *Wood et al., 2003*), could moderate affinity to limit off-DNA polymerization. It is also possible that TEL polymerizes extensively off-DNA and that this polymerization provides a regulatory strategy that moderates the extent of on-DNA binding and repression. It should also be noted that SY5 exhibited sub-nanomolar affinity and migrated slower than the TEL SAM in the native gel assay, and so even though TEL polymerization is quite strong, it is still tuned to sub-maximal affinity.

From a structural perspective, our in vitro results emphasize the importance of charged residues at the periphery of the SAM–SAM interface in determining affinity. Previously, mutation of TEL residues K99, E100, and R103 (Yan residues R92, A93, R96) was shown to reduce both TEL polymerization in vitro and the transforming potential of the TEL-NTRK3 oncogenic fusion protein in vivo (*Cetinbas et al., 2013*). Our results confirm the importance of these three residues to high-affinity polymerization and add Yan H97Y to the suite of residues that control SAM–SAM interactions. In addition, we note that SY5 is the strongest quantitatively measured SAM–SAM interaction (*Bowie and Qiao, 2005*), which suggests evolutionary selection against sub-nanomolar affinities, presumably to prevent excessive self-aggregation.

Last, it is interesting to consider Yan polymerization in the context of the phase separation model for transcriptional control that has been proposed recently (*Hnisz et al., 2017*). Briefly, the assembly of proteins into dynamic aggregates known as membrane-less organelles, or liquid-like droplets, is thought to provide a general regulatory strategy for organizing and partitioning specific biochemical reactions and functions within cells (*Kaganovich, 2017*). It has been proposed that phase separation of TF complexes could underlie many aspects of transcriptional regulation, including super-enhancer formation and function, transcriptional bursting and long-range interactions (*Hnisz et al., 2017*). We speculate that Yan may participate in granule-like complexes, and that SY could have an increased propensity to phase-separate when off-DNA. In support of this idea, nuclear puncta are detected with wildtype Yan, although not nearly as frequently as with SY. Furthermore, low-complexity domains or intrinsically disordered regions are especially important for phase-separation behavior and are frequently enriched in TFs (*Staby et al., 2017*), including Yan (*Lai and Rubin, 1992*). Finally, the description of Yan genome-wide chromatin occupancy reveals patterns that closely resemble those described for super-enhancers (*Webber et al., 2013*; *Whyte et al., 2013*). High-resolution live imaging and biochemistry will be necessary to assess whether the sub-nuclear dynamics of Yan and SuperYan match expectations for phase-separated complexes and how this impacts the context-specific repression of target genes.

## Materials and methods

### Modeling of Yan fractional occupancy

Fractional occupancy of Yan was calculated largely in accordance to previously described methods (*Hope et al., 2017*), which use an equilibrium approach that depends solely on the affinities of protein–DNA and SAM–SAM affinity, as well as on the concentration of Yan and size of DNA element. For all calculations, the size of the element was set to 24 sites, with a single ETS site in the first position, followed by 23 non-specific DNA sites. For clarity, *Figure 1* and *Figure 1—source data 1* show fractional occupancy just at the single ETS site. In addition, the wildtype values of Yan-specific and -non-specific protein–DNA affinity used to calculate Yan occupancy previously ($\alpha_{wt}$ and $\beta_{wt}$) were set to $-9.955$ kcal/mol and $-5.837$ kcal/mol, respectively, for all calculations.

The modification of the model to account for polymerization off of DNA takes the form of a multiplier on the effective concentration of free Yan to bind DNA. The fraction of free Yan monomers is given by $1 / \{1 + [\text{Yan}]/K_d + ([\text{Yan}]/K_d)^2 + ([\text{Yan}]/K_d)^3 + \ldots ([\text{Yan}]/K_d)^n\}$ where [Yan] is the concentration

of total Yan, $K_d$ is the dissociation constant of SAM–SAM affinity, and *n* is the maximum length of the polymer. All calculations were performed at $n = 50$, as increasing *n* was shown to approach asymptotically the same fractional occupancy curves. To calculate the binding of multiple Yan species (e.g. monomers and dimers; monomers, dimers, and trimers; etc.) for *Figure 1—source data 1*, the numerator of the above expression was expanded to include the desired number of species: 1 +[Yan]/Kd for monomers and dimers, and $1+[Yan]/K_d + ([Yan]/K_d)^2$ for monomers, dimers, and trimers. To arrive at the concentration of 50% fractional occupancy, values of Yan concentration were tested empirically for a given SAM–SAM affinity, up to five significant figures.

## Molecular biology and cloning

The DNAs encoding the Yan SAM (S33-S117) and the TEL SAM (A40-Q123) were subcloned into the NotI/HindIII sites of His-A scGFP-30/pBAD (kindly provided by James Bowie and Catherine Leetola). Subsequent generation of point mutations YT1-15, TEL$^{A93D}$, TEL$^{V112E}$, Yan$^{A86D}$ and Yan$^{V105R}$, and of constructs for SPR was carried out by Quikchange Mutagenesis (Stratagene), followed by sequencing. For SPR constructs, either a TEV cleavage site or a 6XHis-tag with a TEV cleavage site was inserted into the KpnI/NotI sites of the relevant YT-negGFP/pBAD construct.

To generate expression constructs for co-immunoprecipitation, full-length YanWT/pENTR3C construct (*Zhang et al., 2010*) was used for LR-mediated exchange into pAWF and pAWH expression vectors. The C-terminal stop codon was removed via Quikchange mutagenesis.

To generate constructs for S2 cell transcription assays, untagged Yan$^{WT}$/pMT was used as a substrate for Quikchange to generate SY1–5/pMT. Yan$^{ACT}$/pMT, RasV12/pMT, and pMT empty have been generated previously (*Zhang et al., 2010*). FLAG–PntP1/pMT was generated by ligating a PCR-amplified PntP1-encoding product into the KpnI/SalI sites of an N-terminal FLAG-pMT vector.

To generate upstream activating site (UAS) constructs for overexpression, the full-length Yan sequence was amplified from Yan/pMT templates and subcloned as a KpnI/XbaI fragment into pUASTattb.

To generate genomic rescue constructs for CRISPR/Cas9-mediated mutagenesis, the Yan locus was amplified using genomic DNA extracted from the strain *w,y,vasa >cas9*, at positions 2L:2,159,158 – 2L:2,161,166 (*Dm* Assembly 6). The resulting ~2 kb product was digested with KpnI/SacI and subcloned into pBS-SKII$^+$, and then mutated by Quikchange to generate SY3-5. To generate the gRNA construct, primers were annealed and ligated into BbsI-digested pU6-gRNA.

To generate the *salm >GFP* reporter, the 5′ region identified on the basis of the Yan ChIP profile was amplified from the genome with primers 5′-GCGCTCGAGACACAAACAATAACAGCCGCTAC-GAATAACAG-3′ and 5′-CTGGCTAGCGCTAAAAATTTCTCATTTGCAGAGAGGCAACG-3′ to produce a 1000 bp fragment from 2L:11,445,615 to 2L:11,446,614, which was ligated into XhoI/NheI digested pJR20. For the *salm >Luciferase* reporter, the same region was amplified with primers 5′-GCGCTGCAGACACAAACAATAACAGCCGCTACGAATAACAG-3′ and 5′-CTGGGTACCGC TAAAAATTTCTCATTTGCAGAGAGGCAACG-3′ and cloned into the PstI/KpnI sites of pBluescript-luciferase.

## Fly strains and genetics

Transgenics of UAS-Yan and UAS-SY1-5 were generated by injecting into *vasa >phiC31;86Fb-attB* (BL#247249). CRISPR/Cas9-mediated mutagenesis was accomplished by injecting into *vasa >Cas9*. Additional strains from the Bloomington stock center: *w$^{1118}$*, *ey-FLP;Ubi-GFP,FRT40A*, *GMR-GAL4*, and *w⁻;Sco/CyO,Dfd-YFP(CDY)*. Yan$^{443}$ and Yan$^{V105R}$ alleles were as described by *Webber et al., 2013*).

To generate UAS transgenes, the *salm* reporter, and the CRISPR/Cas9 mutants, embryos were injected in accordance with the procedure outlined by *Fujioka et al., 2000* using 1 ug/uL for Yan/puASTattB or *salm* reporter constructs or 1 ug/uL of the relevant genomic rescue construct and 50 ng/uL of gRNA for Cas9 mutagenesis. Transgenic flies were identified by expression of the mini-*w$^+$* marker, whereas CRISPR-mediated mutants were screened for enhancement of the *sev +RasV$^{V12}$* rough eye phenotype (*Rebay et al., 2000*) and then confirmed by PCR and sequencing.

## negGFP native gel assay

This assay was adapted from *Knight et al. (2011)*. SY-negGFP/pBAD constructs were transformed into DH10B cells. 200 mL cultures were grown at 37°C for ~2.5 to an OD600 of 0.6 – 0.8 and induced at 18°C with 0.2% L-(+)-arabinose (Sigma) for 8 hr. Cells were pelleted, frozen, and lysed in SAM Native Gel Lysis Buffer (20 mM Tris, 1M NaCl, 5 mM $MgCl_2$, pH 7.5) with 1 mM DTT, 1 mg/mL lysozyme, and complete Mini protease inhibitor tablets (1 tablet per 10mLs of buffer; Roche). Lysis proceeded for 30 min with rocking at 4°C, followed by sonication (Fisher Sonic Dismembrator Model 500, 1 min total time, 10 s on, 10 s off, 20% amplitude). Lysates were cleared by spinning at 16,000 x $g$ at 4°C for 15 min, and then incubated with HisPur Cobalt Resin (Thermo) in Binding Buffer (50 mM sodium phosphate, 300 mM NaCl, 10 mM imidazole, pH7.4) for 60 min, according to the manufacturers instructions. Beads were washed twice with two bed volumes of binding buffer, and then eluted in elution buffer (binding buffer + 150 mM imidazole). The fluorescence of YT constructs was measured with a 96-well plate reader (Synergy Neo HST) diluted 1:10 in elution buffer, with filters set to an excitation wavelength of 485 nm, an emission wavelength of 516 nm, and gain set to 35. Measurements were buffer subtracted and constructs were standardized to 375,00 RFU.

To run the native gel assay, 10-well native acrylamide gels were poured at 12.5% and 6%, with a stacker of 3% acrylamide. Samples were loaded with 5X native gel sample buffer (300 mM Tris, 50% glycerol, bromphenol blue, pH 6.8), and gels were run in native gel running buffer (40 mM Tricine, 60 mM Tris) at 4°C at 22V for ~48 hr (12.5% gels) or ~18 hr (6%). Gels in *Figure 3—figure supplement 1* were run identically, except that gradient gels (Mini-PROTEAN TGX Stain-Free, 4 – 15%, Bio-Rad) were used instead of single-percentage gels, run for ~24 hr. Gels were imaged on a Typhoon imager, with excitation 488 nm and emission of 516 nm.

## SPR purification and measurement of affinity

Proteins prepared as above were cleaved with TEV protease overnight at 4°C at an $A_{280}$ ratio of construct to TEV of 100:1. Constructs were further purified with an AktaPure FPLC system equipped with a HiLoad 16/600 Superdex75 column, in SAM native gel lysis buffer. Samples eluted in two peaks, consistent with the larger negGFP fragment and the smaller SAM domain, and the peak centered on 80 mL of elution volume was collected and concentrated using Amicon Ultra 15 – Ultracel 3K centrifugal concentration filters (Millipore). Samples were flash-frozen in liquid nitrogen using 10% glycerol as cryoprotectant. Residual His-binding was removed via another 1 hr incubation with HisPur Cobalt Resin in situ, and concentrations were measured via $A_{280}$ in triplicate with a NanoDrop ND-1000 Spectrophotometer. SPR measurements were taken with a ProteonXPR36 machine (Bio-Rad) using a HTE chip. 5 mM $NiSO_4$ was used to regenerate the chip, and ligand was used at a concentration of 100 nM for each interaction. Sensograms were measured in duplicate, and then fitted to a two-state kinetic binding model.

## S2 cell culture and transfection

*Drosophila* S2 cells were maintained in Schneider Insect Media (Sigma) with Insect Media Supplement (Sigma) and penicillin/streptomycin. Cells were grown confluence and transfected using 250 ug/mL DDAB (dimethyldidodecylammonium bromide, Sigma) as previously described (*Zhang et al., 2010*).

## Co-immunoprecipitation

Co-immunoprecipitation experiments were performed as described previously (*Zhang et al., 2010*). 1500 ng of the appropriate Yan construct (in pAWF or pAWH expression vectors) were transfected into confluent S2 cells, with $9.0 \times 10^6$ cells in 6 mL of S2 media used for each experiment. Cells were lysed in IP lysis buffer (50 mM Tris, 100 mM NaCl, 1% NP-40, 2 mM EDTA, 2 mM EGTA, pH 8.0) with cOmplete Mini protease inhibitor tablets and 0.5 mM DTT. Cleared lysates were incubated with 20 uL of a 1:1 slurry of Anti-FLAG-M2 Affinity Gel (Sigma) for 2 hr. Samples were washed three times with 500 uL of IP lysis buffer, resuspended in sample buffer and run on an SDS PAGE gel. After transfer to nitrocellulose, the membrane was blocked with 1% casein for 1 hr, incubated with 1:3000 mouse anti-HA and 1:1000 rabbit anti-FLAG overnight at 4°C, washed, incubated with 1:2000 Alexa 488 and 1:2000 Alexa 600 for two hours, washed and imaged.

## Transcription assays

*Drosophila* S2 cells were transfected with 100 ng of *king-tubby*, *salm*, or 6X-ETS luciferase reporter construct, 100 ng of the relevant Yan/pMT construct, 100 ng of PntP1/pMTFLAG, 20 ng of actin >Renilla luciferase, and if applicable, 5 ng of Ras$^{V12}$/pMT. Cells were lysed in 170 uL transcription assay lysis buffer (100 mM potassium phosphate, 0.5% NP-40, pH7.8), and incubated at 4°C for 30 min. $2.25 \times 10^6$ cells in 1.5 mL of S2 media were used for each experiment. Luciferase measurements were made using an Autolumat Plus LB 953, using luciferase buffer (10 mM Mg acetate, 100 mM tris acetate, 1 mM EDTA, pH 7.8) with 4.5 mM ATP (Fisher) and 77 uM D-luciferin (Pierce), and Renilla buffer (25 mM sodium pyrophosphate, 10 mM Na acetate, 15 mM EDTA, 500 mM $Na_2SO_4$ 500 mM NaCl, pH 5.0) with 4 mM coelenterazine (Promega). Luciferase measurements were made in technical triplicates (50 uL per sample) for each biological replicate, and the ratio of Firefly RLU to Renilla RLU was taken as transcriptional activity, and then all measurements were normalized to reporter alone.

## Immunohistochemistry, histology, and microscopy

*Drosophila* S2 cells transfected with 1000 ng of the relevant Yan/pMT construct, were settled on poly-L lysine coated slides for one hour, fixed with 4% para-formaldehyde and 0.1% Triton for 10 min, washed 5X with PBT (PBS + 0.1% Triton), incubated with 1:500 mouse anti-Yan (DSHB 8B12H9) in PBT with 5% normal goat serum (NGS) for 1 hr, washed 5X with PBT, incubated with anti-mouse Cy3 1:2000 and DAPI 1:2000 in PBT with 5% NGS for 1 hr, washed 5X with PBT and mounted with n-propyl gallate mounting medium. Approximately 200 cells were counted for each genotype.

White pre-pupal third instar larvae were dissected in S2 cell medium and eye discs were fixed, blocked, and incubated as above, except that primary antibody incubation was at 4°C overnight. Antibodies: mouse anti-Yan monoclonal antibody at 1:500 (DSHB 8B12H9), guinea pig anti-salm polyclonal antibody at 1:500 (a gift from Claude Desplan), and mouse anti-Pros monoclonal antibody at 1:50 (DSHB MR1A). Secondary antibodies: Cy3 conjugated goat anti-mouse (Jackson Immunoresearch) and anti-guinea pig (Jackson Immunoresearch) and DAPI, all at 1:2000. Confocal images were taken on a Zeiss LSM 880 microscope, using 0.5 um to 1.0 um slices. Quantification of Yan vs. SY4 levels was done by outlining clones of comparable size, using the absence of GFP to mark SY5 tissue and the presence of GFP to mark wildtype Yan tissue. Total fluorescence was measured by counting integrated pixel intensity within the clonal boundary (ImageJ Tools). DAPI fluorescence was used to create a mask of nuclei within clones, and nuclear fluorescence was measured in the same way for the regions that fell inside both the clone and the nuclear mask.

To image adult eyes, decapitated heads were imaged with a Canon EOS Rebel camera fitted to a Leica dissecting microscope. Adult eye histology and sectioning was performed as described previously (*Davis et al., 2017*), except that embedded heads were incubated in 100% resin for 24 hr at room temperature.

## Chromatin immunoprecipitation and sequencing

Eye imaginal discs were dissected from late third instar larvae in batches of 30 – 50 pairs. Discs were placed in 1 ml of S2 cell medium and kept on ice. Samples were then incubated in 1 ml of cross-linking solution (50 mM HEPES, pH 7.6, 1 mM EDTA, 0.5 mM EGTA, 100 mM NaCl, 1.8% formaldehyde) for 15 min with rocking, followed by Stop solution (PBS, 0.1% Triton X-100, 125 mM glycine) for 5 min with rocking. After washing with PBT (PBS, 0.1% Triton X-100) discs were homogenized in ChIP lysis buffer (50 mM HEPES, pH 7.6, 140 mM NaCl, 1 mM EDTA, 1 mM EGTA, 1% Triton X-100, 0.1% sodium deoxycholate, protease inhibitor tablet (Roche)). Samples were then stored frozen at −20°C. Once a total of 130 discs per genotype had been achieved, samples were pooled and chromatin was sonicated to approximately 300 – 500 bp using a Fisher Scientific Sonic Dismembrator sonicator (model 500) with nine cycles at 15% amplitude for 15 s (0.9 s on/0.1 s off). Clarified lysates were incubated with guinea-pig anti-Yan overnight at 4°C. Gamma-bind sepharose beads were added and incubated for 4 hr at 4°C. Beads were washed in ChIP lysis buffer, high-salt ChIP lysis buffer (ChIP lysis buffer with NaCl adjusted to 500 mM) and TE (10 mM Tris, pH 8, 1 mM EDTA) and then resuspended in TE/SDS (10 mM Tris, pH 8, 1 mM EDTA, 1% SDS) and reverse cross-linked at 65°C overnight. ChIP DNA was purified using a PCR purification kit (MIDSCI Scientific).

ChIP signals were quantified using the QuantiTech SYBR Green PCR Kit (QIAGEN). Standard curves were generated for each primer pair using serial dilutions of genomic DNA. Relative amounts of input and immunoprecipitated DNA were determined on the basis of the standard curves, and the ChIP signals calculated as IP/input ratios. Reaction conditions were 95°C for 30 s, 55°C for 30 s and 72°C for 30 s (45 cycles).

Primers and probes used for qPCR were as follows: *spalt* promoter — Forward ACT CCC TCT CTC TCT TTC TCT C, Reverse AAC AAC AAT GGC GCA AAG G; *prospero* enhancer — Forward AGG GTT TCG AGT TGC CTT AAT, Reverse ACA CAC CTT TGT TTG CCT TTG; *aos* — ForwardTGA ATA CGC TGC AGT TTA AG, Reverse AAC TGA CGG AGG AAG TAA ATA A; *aop* —Forward CTC ATG AGT ATA CCC AGC AAT, CTA AAT GGG ACG TAA GGT TG; N.C. — Forward GCA TTT ATT AAG GCC AAC AC,

Reverse GTT AAG CTT AGG TCG TGC TC.

For ChIP-seq, two biological replicates of ChIP and an input sample were sequenced on an Illumina Genome Analyzer. The Eland program was used to align sequence reads to the Dm3 *Drosophila melanogaster* genome and the alignments were converted into a wiggle file using the ChIP-seq R package SPP (*Kharchenko et al., 2008*). Input data were subtracted and rescaled from the IP data to generate final tag density wig files for visualization of peaks in the genome browser IGB (*Nicol et al., 2009*). Full results will be reported elsewhere.

## Statistics

No explicit power analysis was used to pre-determine sample sizes. Sample sizes were determined on the basis of previous experiments that had shown significance (for example, transcription assays and lethality assays as performed; *Webber et al., 2013*). Numbers of replicates for individual experiments are noted in the corresponding figures and legends. For transcription assays, separate transfections were taken to be biological replicates, whereas repeated measurements of the same cell lysate were taken to be technical replicates. Outliers were not discarded from analysis. Experiments were not randomized, but S2 cell sub-cellular localization was scored in a blinded fashion.

## Acknowledgements

The authors thank James Bowie, Catherine Leetola, and Claude Desplan for reagents, John Reinitz and Kenneth Barr for modeling advice, James Fuller for biochemical insight and purification help, Juana Delao for analysis of *Drosophila* retinas, Jean-Francois Boisclair Lachance for the Pros-GFP reporter, Elena Solomaha and the University of Chicago Biophysics Core for assistance with SPR measurements, and Michael Glotzer, Kenneth Barr, James Fuller, and members of the Rebay lab including Nicelio Sanchez-Luege and Trevor Davis for helpful discussion and critical reading of the manuscript. We acknowledge the Bloomington Drosophila Stock Center (NIH P40OD018537) and the Developmental Studies Hybridoma Bank (created by the NICHD of the NIH) for critical reagents. MH, JLW, and IR are supported by NIH grant R01 GM080372. The work was also supported by the Genomics Core Facility through a University of Chicago Cancer Center Support Grant P30 CA014599. MH has been supported by NIH grants T32 GM007183 and 2T32 HL007381-36A1. JLW was supported by American Heart Association grants 12POST12040225 (2012 – 2014) and 15POST22660028 (2015). SAT. was supported by NSF REU 1659490.

## Additional information

### Funding

| Funder | Grant reference number | Author |
| --- | --- | --- |
| National Institutes of Health | R01 GM080372 | Ilaria Rebay |
| National Science Foundation | REU 1659490 | Sherzod A Tokamov |
| National Institutes of Health | P30 CA014599 | Ilaria Rebay |
| National Institutes of Health | T32 GM007183 | C Matthew Hope |
| National Institutes of Health | 2T32 HL007381-36A1 | C Matthew Hope |

| American Heart Association | 12POST12040225 | Jemma Webber |
| American Heart Association | 15POST22660028 | Jemma Webber |

The funders had no role in study design, data collection and interpretation, or the decision to submit the work for publication.

### Author contributions
C Matthew Hope, Conceptualization, Software, Formal analysis, Supervision, Validation, Investigation, Visualization, Methodology, Writing—original draft, Project administration, Writing—review and editing, Designed and performed experiments, Wrote the modeling code, Analyzed data; Jemma L Webber, Investigation, Writing—review and editing, Performed experiments, Analyzed data; Sherzod A Tokamov, Investigation, Performed experiments; Ilaria Rebay, Conceptualization, Formal analysis, Supervision, Funding acquisition, Investigation, Visualization, Methodology, Writing—original draft, Project administration, Writing—review and editing, Designed and performed experiments, Analyzed data

### Author ORCIDs
Ilaria Rebay (iD) http://orcid.org/0000-0002-2444-3864

### Decision letter and Author response
Decision letter https://doi.org/10.7554/eLife.37545.024
Author response https://doi.org/10.7554/eLife.37545.025

## Additional files

### Supplementary files
• Transparent reporting form
DOI: https://doi.org/10.7554/eLife.37545.019

### Data availability
A published ChIP dataset was used in this study: Webber JL, Zhang J, Cote L, Vivekanand P, Ni X, Zhou J, Nègre N, Carthew RW, White KP, Rebay I. Genetics. 2013. The relationship between long-range chromatin occupancy and polymerization of the Drosophila ETS family transcriptional repressor Yan. Raw data for this published study are available as a GEO dataset (Series: GSE34038 and GSE34040). All other data analysed during this study are included in the manuscript and supporting files. Source data files have been provided for Figures 1 and 3.

The following previously published datasets were used:

| Author(s) | Year | Dataset title | Dataset URL | Database and Identifier |
|---|---|---|---|---|
| Webber JL, Zhang J, Cote L, Vivekanand P, Ni X, Zhou J, Nègre N, Carthew RW, White KP, Rebay I | 2013 | The relationship between long-range chromatin occupancy and polymerization of the Drosophila ETS family transcriptional repressor Yan | https://www.ncbi.nlm.nih.gov/geo/query/acc.cgi?acc=GSE34038 | NCBI Gene Expression Omnibus, GSE34038 |
| Webber JL, Zhang J, Cote L, Vivekanand P, Ni X, Zhou J, Nègre N, Carthew RW, White KP, Rebay I | 2013 | The relationship between long-range chromatin occupancy and polymerization of the Drosophila ETS family transcriptional repressor Yan | https://www.ncbi.nlm.nih.gov/geo/query/acc.cgi?acc=GSE34040 | NCBI Gene Expression Omnibus, GSE34040 |

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
