## [Decision Letter]

Thank you for submitting your article "Tuned polymerization of the transcription factor Yan limits off-DNA sequestration to confer context-specific repression" for consideration by *eLife*. Your article has been reviewed by three peer reviewers, and the evaluation has been overseen by Philip Cole as the Reviewing and Senior Editor. The following individual involved in review of your submission has agreed to reveal his identity: Steve Gisselbrecht (Reviewer #1). Two other reviewers remain anonymous.

The reviewers have discussed the reviews with one another and the Reviewing Editor has drafted this decision to help you prepare a revised submission.

Summary:

This is an interesting case study of a polymerizing transcription factor (Yan) in a developmental system with relevance to human cancer (TEL), where the function of monomeric, dimeric, higher-order polymer filaments is unknown. The authors create Super-Yan mutants (SY) with increasing protein-protein affinity to characterize the biochemistry, gene dosage effects (homo/heterozygote, overexpression), and phenotypes of different Yan alleles. The data in this paper are of high quality and the conclusions are well justified by the data. The paper brings out some interesting ideas about the role of protein-protein interactions in regulating binding site occupancy and specificity.

Overall, it was felt that the authors really should integrate mathematical modeling with their data to address Yan mechanism and function. Their previous model (Hope et al., 2017) suggested that Yan polymerization on DNA better repressed transcription (thus, higher affinity equals stronger repression). Here, they consider an alternative where cytoplasmic Yan polymer sequesters monomers away from DNA (thus, higher affinity equals weaker repression). Each mechanism alone or in combination will exhibit different quantitative responses to allele concentration and affinities, which the authors have measured. It should be possible for the authors to update their model to include multiple Yan alleles (diploid genetics) and measured protein-protein affinities (k_on_, k_off_, K_d_) of these alleles. Their model, if correct, should be able to quantitatively explain the collective data (monomeric Yan, wild-type Yan, Super-Yan alleles in homo-/ heterozygote, overexpression strains).

Essential revisions:

1) The hypomorphic and recessive nature of SY alleles (despite a 10000x increase in SY-SY affinity) is surprising, which suggests the current model (Hope et al., 2017) is incomplete or wrong. What is the affinity of WT and SY allele for each other? What happens to Yan polymerization and repression in a heterozygote in the mathematical model and experiments?

2) The modeling relies on an assumption about which oligomer species can bind DNA. The sentence in the second paragraph of the subsection “Modeling Yan DNA occupancy predicts a requirement for tuned protein-protein interaction affinities to limit off-DNA aggregation”, which states that the assumption that only monomers can find their DNA binding sites is supported by the observation that monomeric species bind substantially the same sites as those that can oligomerize, does not seem to obviously follow. Can the model address this monomer binding assumption and the observation that monomeric Yan species bind the same sites as wild-type Yan that can polymerize, but not repress?

3) A more detailed explanation of the assumptions that went into the equilibrium modeling is needed. It is not obvious why there should be both an upper and a lower bound on the concentration of Yan that will allow 50% or greater binding site occupancy and why both of these bounds should be dependent on SAM-SAM affinity. Perhaps graphs of binding site occupancy as a function of Yan concentration for different SAM-SAM affinities would be useful.

4) In addition to showing the Yan SAM-SAM affinity on the graph in Figure 1, it would be interesting to see the where the SAM-SAM affinities for the various SuperYan mutants fall (especially the three that were used in the genetic analysis).

5) The authors suggest two different models to explain the differential effects of monomeric and SuperYan mutants on R3/4 vs. R7 fate. In the second model, they propose that different levels of Ras signaling in the two cell types results in differential ability to disrupt Yan oligomerization. Is there, in fact, evidence that Ras signaling disrupts Yan oligomerization? Is this something that the authors could examine, perhaps using the S2 cell system in conjunction with the SuperYan mutants?

6) Are there candidate Yan target genes responsible for the ability of Yan to suppress the R3/4 fate and other target genes responsible for the ability of Yan to suppress the R7 fate. If so, would it be possible to test the predictions that the authors make regarding the effect of SuperYan mutations on the occupancy of these genes either by in silico modeling or by Yan ChIP?

7) The vast majority of transcription factors do not contain polymerization domains such as the SAM domain. Do the authors believe that the findings presented here about tuned protein-protein interactions are only relevant to the small subset of factors that contain polymerization domains? Or do they believe that the findings can be generalized to other types of transcription factors? This is something that should be discussed.

---

## [Author Response]

Essential revisions:1) The hypomorphic and recessive nature of SY alleles (despite a 10000x increase in SY-SY affinity) is surprising, which suggests the current model (Hope et al., 2017) is incomplete or wrong. What is the affinity of WT and SY allele for each other? What happens to Yan polymerization and repression in a heterozygote in the mathematical model and experiments?

The reviewers are correct that the SuperYan loss-of-function phenotypes cannot be explained by the original model described in Hope et al., 2017. The key assumption of that original model was that Yan polymerization only occurs on DNA. Using that simple premise, higher affinity was predicted to promote Yan occupancy and repression, in essence producing gain-of-function effects. By considering only DNA-bound complexes, the original model ignored the non-DNA bound pool of transcription factor from which regulatory DNA-bound complexes are assembled. In this paper we present a revised model that also considers off-DNA polymerization.

To clarify the differences in the predictions that these two models make, we have expanded Figure 1. Panels 1A-C have been added to compare the predictions of the two models under wild type, weak or strong SAM affinity. The key difference emerges when considering strong SAM affinity (1C). While the original model predicts high occupancy, the new model predicts the opposite, namely a loss of DNA occupancy, and hence loss-of-function. These polar opposite predictions set the perfect stage for the experimental analysis presented in the rest of the paper. In brief, the project was designed to test which model better recapitulates the actual biology. The results are unequivocal – only the new model that has been updated to consider Yan polymerization both on and off of DNA can explain the loss of function behavior of the SuperYan alleles. The striking increase in Yan puncta observed in SuperYan discs appears consistent with the idea that stronger SAM affinity will promote excessive aggregation, thereby precluding the formation of effective transcriptional complexes, although formal proof of the subcellular basis of the SuperYan phenotypes remains a non-trivial task for the future. However, even though the old model does not predict the SuperYan loss-of-function effect, some of its conclusions may still be relevant in considering mechanisms of Yan occupancy and repression at different enhancers. To discuss these points more thoroughly, two new paragraphs have been added to the Discussion (tenth and eleventh paragraphs).

The reviewers also commented that the recessive nature of the SuperYan alleles was surprising and suggested we consider whether the affinity of the wild type Yan-SuperYan interaction predicts this behavior. To address this point, we first confirmed that Yan and SuperYan can associate in common complexes by performing heterotypic co-immunoprecipitation experiments between wild type Yan and the strongest SuperYan allele SY5. These new results are shown in Figure 3—figure supplement 3 and are described in the last paragraph of the subsection “A screen for Yan mutants that increase SAM-SAM affinity”. Second, we examined puncta in *yan/yan^SY5^* discs and found although there is measurable 50% increase in the number of puncta per area relative to wild type there is much larger (~8x) reduction relative to *yan^SY5^* homozygotes (these new data are shown as Figure 5—figure supplement 1 and described in the last paragraph of the subsection “High affinity SAM-SAM interactions promote nuclear and cytoplasmic Yan aggregation”. Assuming puncta reflect SAM-SAM driven aggregates, this argues that wild type Yan effectively interacts with and interrupts SY5 polymers to support wild type function, thereby at least partly explaining why only SY homozygotes have obvious phenotypes. Third, although attempting direct biochemical measurements was beyond the scope of what we felt we could accomplish in the revision period, it is possible to make inferences about Yan-SY affinity from our existing SPR data set because the affinities of WT to SY5 at either the EH or ML surface will differ from our measurements by one mutation. We remind the reviewers that three of the mutations in SY5 (R92K, G96R, and H97Y) are found at the EH surface, while the fourth (A93E) is found at the ML surface. Since the measurement of SY3 affinity as ~10nM has two of the relevant mutations for the WT-SY5 interaction (R92K and G96R) but lacks the third (H97Y), we expect the WT-SY5 interaction to be at least 10nM. Conversely at the ML surface, our measurement of the SY2 interaction has one relevant mutation (A93E) and one extra mutation (G96R), so we expect WT-SY5 affinity to be weaker than the ~60nM measured for SY2. Because of its complexity and detail, we have not included this third point in the revised manuscript, but if the reviewers feel it should be added, we are happy to do so.

Finally the reviewers suggested that we include heterotypic interaction between Yan and SuperYan in our model. While at one level this would be “easy” to do, the reality is that our current calculations are already straining the memory limit of our computing resources. Consideration of heterotypic and homotypic interactions on DNA would triple the number of microstates to be calculated, making it prohibitively expensive with current computing resources. Thus while we strongly concur with the reviewers that future models must incorporate heterotypic interactions (not just Yan-SY, but also with other transcription factors) doing that will require substantial algorithmic changes to how DNA occupancy is calculated, something we did not feel was a feasible short-term goal.

2) The modeling relies on an assumption about which oligomer species can bind DNA. The sentence in the second paragraph of the subsection “Modeling Yan DNA occupancy predicts a requirement for tuned protein-protein interaction affinities to limit off-DNA aggregation”, which states that the assumption that only monomers can find their DNA binding sites is supported by the observation that monomeric species bind substantially the same sites as those that can oligomerize, does not seem to obviously follow. Can the model address this monomer binding assumption and the observation that monomeric Yan species bind the same sites as wild-type Yan that can polymerize, but not repress?

The reviewers request clarification of our assumptions regarding which oligomer species can bind DNA. We have done so (subsection “Modeling Yan DNA occupancy predicts a requirement for tuned protein-protein interaction affinities to limit off-DNA aggregation”, second paragraph). Briefly, as described by the Stokes-Einstein relation, the inverse relationship between particle size and diffusion size means that as Yan oligomers increase in length, diffusion and DNA binding will be reduced and eventually prevented. To elaborate, the assumption that smaller Yan species will be free to diffuse and bind their targets on DNA stems from the equation of 3D diffusion of small particles. The root mean square displacement of a particle is inversely proportional to its size. Therefore as the size of a Yan polymer complex grows larger, the rate of diffusion will shrink until the complex is so large as to prevent Brownian diffusion. We assume this would also preclude binding to target DNA. We do not have an empirical measurement as to what size of Yan polymer would be unable to diffuse, but Figure 1—figure supplement 1 shows that if species including monomers, dimers, and trimers are allowed to bind DNA, the results are not substantially different from monomers alone.

In contrast to Stokes-Einstein based assumptions, the observation that Yan monomers bind the same regions of DNA as wild type Yan, but do not repress effectively, is an empirical one based on genome-wide ChIP data and phenotypic analysis (Webber et al., Genetics 2013) (subsection “Modeling Yan DNA occupancy predicts a requirement for tuned protein-protein interaction affinities to limit off-DNA aggregation”, last paragraph).

3) A more detailed explanation of the assumptions that went into the equilibrium modeling is needed. It is not obvious why there should be both an upper and a lower bound on the concentration of Yan that will allow 50% or greater binding site occupancy and why both of these bounds should be dependent on SAM-SAM affinity. Perhaps graphs of binding site occupancy as a function of Yan concentration for different SAM-SAM affinities would be useful.

The reviewers recommended we add graphs of occupancy as a function of Yan concentration for different SAM-SAM affinities to clarify the equilibrium modeling results. We have done so. Figure 1A-C now present fractional occupancy curves that clearly show a rise in Yan occupancy, saturation, and then decrease in Yan occupancy, as a function of Yan concentration and SAM-SAM affinity. The results are also described more extensively (subsection “Modeling Yan DNA occupancy predicts a requirement for tuned protein-protein interaction affinities to limit off-DNA aggregation”, third paragraph).

4) In addition to showing the Yan SAM-SAM affinity on the graph in Figure 1, it would be interesting to see the where the SAM-SAM affinities for the various SuperYan mutants fall (especially the three that were used in the genetic analysis).

As suggested, we have marked the affinity of the five SuperYan mutants SY1-5 on the graph in Figure 1D.

5) The authors suggest two different models to explain the differential effects of monomeric and SuperYan mutants on R3/4 vs. R7 fate. In the second model, they propose that different levels of Ras signaling in the two cell types results in differential ability to disrupt Yan oligomerization. Is there, in fact, evidence that Ras signaling disrupts Yan oligomerization? Is this something that the authors could examine, perhaps using the S2 cell system in conjunction with the SuperYan mutants?

The reviewers asked us to test whether increased Ras signaling disrupts Yan oligomerization. To do so, we overexpressed an activated allele of the *Drosophila* MAPK, *rolled*, known as *Sevenmaker (rl^SEM^*) throughout the developing *yan^SY5^* eye and then quantified the number of puncta per area. A striking 10-fold decrease relative to *yan^SY5^*alone was measured, with the number of puncta almost indistinguishable from that measured in a wild type background. This result suggests that the signaling environment can influence the extent of SAM-driven Yan polymerization. These new data are included in Figure 5—figure supplement 1C, D and described in the last paragraph of the subsection “High affinity SAM-SAM interactions promote nuclear and cytoplasmic Yan aggregation”.

6) Are there candidate Yan target genes responsible for the ability of Yan to suppress the R3/4 fate and other target genes responsible for the ability of Yan to suppress the R7 fate. If so, would it be possible to test the predictions that the authors make regarding the effect of SuperYan mutations on the occupancy of these genes either by in silico modeling or by Yan ChIP?

The reviewers asked whether there are R3/R4-specific Yan targets where we could measure occupancy directly by ChIP. We put significant effort into addressing this point, and have added extensive new data that strongly bolster our argument of context-specific requirements for different Yan polymers. Briefly, we identified *salm*, a known regulator of R3/R4 specification, as a novel direct Yan target, and showed by ChIP that occupancy is reduced in *yan^SY5^*discs and that a *salm>GFP* reporter transgene (that we generated) is derepressed in *yan^SY5^*. Control ChIP experiments showed wild type SY5 occupancy at target genes like *argos* and *aop* that are expected to be involved in most Egfr-mediated cell fate decisions, arguing strongly for context-specificity to Yan repressive complexes at different target genes. Finally, although we were unable to detect a ChIP signal at the known R7-specific Yan target *pros*, we showed that in contrast to *salm>GFP,* a *pros>GFP* reporter was not significantly derepressed in *yan^SY5^* mutant discs. These new data are shown in Figure 7 and Figure 7—figure supplements 1 and 2 and described in a new Results section titled “Spalt major is a direct target of Yan, and is preferentially sensitive to increased Yan polymerization”. The ChIP experiments were performed by Jemma Webber, who has been added as an author.

7) The vast majority of transcription factors do not contain polymerization domains such as the SAM domain. Do the authors believe that the findings presented here about tuned protein-protein interactions are only relevant to the small subset of factors that contain polymerization domains? Or do they believe that the findings can be generalized to other types of transcription factors? This is something that should be discussed.

The reviewers asked us to discuss the general relevance of our findings to other types of transcription factors. We have done so (Discussion, third paragraph).